# Chemical composition and source attribution of submicron aerosol particles in the summertime Arctic lower troposphere

Franziska Köllner[1,a], Johannes Schneider[1], Megan D. Willis[2,b], Hannes Schulz[3], Daniel Kunkel[4],
Heiko Bozem[4], Peter Hoor[4], Thomas Klimach[1], Frank Helleis[1], Julia Burkart[2,c], W. Richard Leaitch[5],
Amir A. Aliabadi[5,d], Jonathan P. D. Abbatt[2], Andreas B. Herber[3], and Stephan Borrmann[4,1]

[1]Max Planck Institute for Chemistry, Mainz, Germany
[2]Department of Chemistry, University of Toronto, Canada
[3]Alfred Wegener Institute for Polar and Marine Research, Bremerhaven, Germany
[4]Institute for Atmospheric Physics, Johannes Gutenberg University Mainz, Germany
[5]Environment and Climate Change Canada, Toronto, Canada
[a]now at: Institute for Atmospheric Physics, Johannes Gutenberg University Mainz, Germany
[b]now at: Department of Chemistry, Colorado State University, Fort Collins, CO, USA
[c]now at: Aerosol Physics and Environmental Physics, University of Vienna, Austria
[d]now at: Environmental Engineering Program, University of Guelph, Guelph, Canada
**Correspondence:** Franziska Köllner (f.koellner@mpic.de)

**Abstract.** Aerosol particles impact the Arctic climate system, both directly and indirectly by modifying cloud properties, yet our understanding of their vertical distribution, chemical composition, mixing state, and sources in the summertime Arctic is incomplete. In-situ vertical observations of particle properties in the high Arctic, combined with modeling analysis on source attribution are in short supply, particularly during summer. We thus use airborne measurements of aerosol particle composition to demonstrate the strong contrast between particle sources and composition within and above the summertime Arctic boundary layer. In-situ measurements from two complementary aerosol mass spectrometers, the ALABAMA and the HR-ToF-AMS, with black carbon measurements from an SP2 are presented. Particle composition analysis was complemented by trace gas measurements, satellite data, and air mass history modeling to attribute particle properties to particle origin and air mass source regions. Particle composition above the summertime Arctic boundary layer was dominated by chemically aged particles, containing elemental carbon, nitrate, ammonium, sulfate, and organic matter. From our analysis, we conclude that the presence of these particles was driven by transport of aerosol and precursor gases from mid-latitudes to Arctic regions. Particularly, elevated concentrations of nitrate, ammonium, and organic matter coincided with time spent over vegetation fires in northern Canada. In parallel, those particles were largely present in high CO environments ($>90\,\mathrm{ppb_v}$). Additionally, we observed that the organic-to-sulfate ratio was enhanced with increasing influence from these fires. Besides vegetation fires, particle sources in mid-latitudes further include anthropogenic emissions in Europe, North America, and East Asia. The presence of particles in the Arctic lower free troposphere, particularly sulfate, correlated with time spent over populated and industrial areas in these regions. Further, the size distribution of free tropospheric particles containing elemental carbon and nitrate was shifted to larger diameters compared to particles present within the boundary layer. Moreover, our analysis suggests that organic matter, when present in the Arctic free troposphere, can partly be identified as low-molecular weight dicarboxylic

acids (oxalic, malonic, and succinic acid). Particles containing dicarboxylic acids were largely present when the residence time of air masses outside Arctic regions was high. In contrast, particle composition within the marine boundary layer was largely driven by Arctic regional processes. Air mass history modeling demonstrated that alongside primary sea spray particles, marine-biogenic sources contributed to secondary aerosol formation by trimethylamine, methanesulfonic acid, sulfate, and other organic species. Our findings improve our knowledge of mid-latitude and Arctic regional sources that influence the vertical distribution of particle chemical composition and mixing state in the Arctic summer.

*Copyright statement.* TEXT

## 1 Introduction

In the face of rapid climate changes in the Arctic (IPCC, 2014, 2018), polar research has intensified to understand the key processes driving these changes and their effects on the Arctic environment (e.g., Serreze and Francis, 2006; Serreze et al., 2009; Flanner et al., 2011; Stroeve et al., 2012; Pithan and Mauritsen, 2014). Observed amplified Arctic warming in recent decades and coupled rapid sea ice retreat are attributed to various causes, including increases in greenhouse gases and local feedback mechanisms, such as the snow/ice-albedo feedback (e.g., Curry et al., 1995; Holland and Bitz, 2003; Serreze and Barry, 2011; Flanner et al., 2011; Stroeve et al., 2012; Pithan and Mauritsen, 2014; Law et al., 2014; Arnold et al., 2016). However, one important aspect concerning the Arctic climate system pertains to the coupling between aerosol, clouds, and radiation. Aerosol particles, as short-lived climate forcers, perturb the Arctic radiative balance by reflecting and absorbing shortwave radiation (aerosol-radiation interaction) and modifying cloud formation and properties (aerosol-cloud interaction). The coupling between aerosol particles, clouds, and radiation strongly depends on particle chemical composition and mixing state (e.g., Haywood and Boucher, 2000; Boucher et al., 2013). Light-absorbing aerosol, such as black carbon (BC) and mineral dust, can impact the regional Arctic climate by a combination of different aerosol-radiation processes. First, light-absorbing particles when deposited on snow and ice can lead to increased sea ice melt by reducing the snow/ice albedo (e.g., Hansen and Nazarenko, 2004; Flanner, 2013; Jiao et al., 2014; Schacht et al., 2019). Second, the presence of absorbing aerosol in Arctic tropospheric layers can lead to warming in the lower troposphere, but to cooling of the surface beneath by absorbing incoming solar radiation in this layer (e.g., Treffeisen et al., 2005; Engvall et al., 2009; Flanner, 2013). The result is an increase in tropospheric stability (e.g., Flanner, 2013). Third, increased radiative forcing by black carbon in mid-latitudes leads to increasing meridional temperature gradient and subsequent increased northward heat transport (e.g., Sand et al., 2013a, b). However, recent studies show that the magnitude of aerosol-radiation interactions of BC is largely dependent on the particle mixing state (Bond et al., 2006; Kodros et al., 2018). In contrast to light-absorbing aerosol, scattering aerosol species, such as sulfate, of both anthropogenic and biogenic origin, exert a net negative shortwave radiative forcing on the Arctic surface by reflecting radiation back to space (Quinn et al., 2008; Yang et al., 2018). Overall, modeling studies focusing on aerosol-radiation interactions demonstrate that reductions in Arctic anthropogenic aerosol (mainly BC and sulfate) likely contributed to

the observed Arctic surface warming in recent decades (e.g., Shindell and Faluvegi, 2009; Najafi et al., 2015; Acosta Navarro et al., 2016; Breider et al., 2017).

Aerosol particles, serving as nuclei for water condensation or nucleation of the ice phase, are fundamental to cloud formation (e.g., Lohmann and Feichter, 2005). The effects of these particles on clouds are important drivers of the Arctic surface energy budget (e.g., Lubin and Vogelmann, 2006; Zamora et al., 2016). It is known that the net radiative effect of Arctic low-level clouds varies significantly with season (Intrieri et al., 2002; Shupe and Intrieri, 2004). Arctic low-level clouds warm the Arctic surface through most of the year. However, for a short period in summer when the incoming solar radiation is maximum over regions with a low albedo, clouds exert a negative radiative forcing on the Arctic surface (Intrieri et al., 2002; Shupe and Intrieri, 2004). Particularly, in the summertime pristine Arctic environment with low aerosol concentrations, cloud formation and properties are sensitive to the available cloud nuclei concentration (Mauritsen et al., 2011; Moore et al., 2013; Leaitch et al., 2016). Particle sources, formation, and atmospheric processing control particle composition, mixing state, and size distribution, which in turn significantly impact the ability of particles to act as cloud nuclei (e.g., Junge and McLaren, 1971; Haywood and Boucher, 2000; Dusek et al., 2006; McFiggans et al., 2006; Moore et al., 2011; Martin et al., 2011; Boucher et al., 2013). In the summertime pristine Arctic boundary layer (BL), marine emissions can contribute significantly to low-level cloud nuclei concentrations (Orellana et al., 2011; Leaitch et al., 2013; Willis et al., 2016; Burkart et al., 2017; Dall'Osto et al., 2017; Baccarini et al., 2020), whereas the episodic transport of anthropogenic and biomass burning aerosol from southern latitudes can have a crucial impact on cloud formation and cloud properties (e.g., Moore et al., 2011; Zamora et al., 2016; Coopman et al., 2018; Norgren et al., 2018). Along with long range transport and aerosol aging processes, oxygenation of the organic aerosol enhances aerosol's hygroscopicity and thus the ability of the particles to form cloud droplets (e.g., Furutani et al., 2008; Jimenez et al., 2009; Chang et al., 2010; Moore et al., 2011). Together, it is important to know the particle composition, mixing state, and size distribution as well as related sources and formation processes to accurately predict the impact of aerosol on the Arctic climate system.

In Arctic summer, organic matter significantly contributes to submicron aerosol mass (e.g., Schmale et al., 2011; Chang et al., 2011; Lathem et al., 2013; Breider et al., 2014; Willis et al., 2017; Leaitch et al., 2018; Lange et al., 2018; Tremblay et al., 2019). Organic aerosol encompasses a large variety of chemical compounds that vary significantly across Arctic sites, owing to differences in sources and chemical processing (e.g., Shaw et al., 2010; Fu et al., 2013; Hansen et al., 2014; Leaitch et al., 2018). Boreal fires and to a lesser extent anthropogenic activities in North America and northern Eurasia can strongly influence the organic aerosol burden in the summer Arctic free troposphere (FT) (Hirdman et al., 2010; Schmale et al., 2011; Matsui et al., 2011a; Lathem et al., 2013; Breider et al., 2014). Whereas in the summertime Arctic BL, organic matter can also be influenced by both primary emissions and secondary chemical processes from Arctic marine and terrestrial sources (e.g., Willis et al., 2016, 2017; Köllner et al., 2017; Croft et al., 2019). Sources and identities of secondary organic aerosol (SOA) are poorly characterized (Willis et al., 2018). However, tracers of SOA, including a oxygenated species such as dicarboxylic acids (DCA) (e.g., oxalic, malonic, and succinic acid) have been detected at Arctic sites (e.g., Kawamura et al., 1996; Kerminen et al., 1999; Fu et al., 2009; Kawamura et al., 2012; Fu et al., 2013; Hansen et al., 2014). Several studies demonstrated that particulate DCA are less abundant in summer than in spring, driven by diminished long range transport of precursors and efficient aerosol

wet removal. In parallel, natural regional sources of DCA become more important as sea ice melts and biological productivity increases (e.g., Kawamura et al., 1996; Kerminen et al., 1999; Kawamura et al., 2010, 2012). DCA are highly water-soluble, thus, the presence of particulate DCA can result in a more hygroscopic aerosol population, which can affect cloud formation and cloud properties (e.g., Giebl et al., 2002; Ervens et al., 2004; Abbatt et al., 2005; Chang et al., 2007). Besides the importance for aerosol-cloud interactions, little is known on the vertical distribution of DCA in Arctic aerosol and the related sources.

Sulfate concentrations in the summertime Arctic FT are largely influenced by anthropogenic sources in northern Eurasia, North America, and East Asia (Shindell et al., 2008; Hirdman et al., 2010; Kuhn et al., 2010; Bourgeois and Bey, 2011; Schmale et al., 2011; Matsui et al., 2011a; Breider et al., 2014; Yang et al., 2018; Sobhani et al., 2018). Whereas in the summertime Arctic BL, sulfate concentrations are dominated by emissions of dimethylsulfide from Arctic marine sources (e.g., Breider et al., 2014; Yang et al., 2018). Sulfur emissions from volcanoes or Smoking Hills in northern Canada may episodically impact Arctic surface concentrations (e.g., Radke and Hobbs, 1989; Breider et al., 2014; Leaitch et al., 2018).

The summertime Arctic BC burden is mainly controlled by vegetation fires, whereas anthropogenic sources contribute less to the overall transport of BC to Arctic regions in summer (Bourgeois and Bey, 2011; Stohl et al., 2013; Breider et al., 2014; Xu et al., 2017; Winiger et al., 2019; Zhu et al., 2020). Nevertheless, anthropogenic BC sources in northern Eurasia can have important contributions to Arctic near surface concentrations, whereas South Asian BC layers are present in the Arctic middle and upper troposphere (Singh et al., 2010; Huang et al., 2010; Bourgeois and Bey, 2011; Sobhani et al., 2018). Regarding high-latitude anthropogenic sources, recent studies demonstrated oil/gas extraction activity and shipping to significantly impact the lower tropospheric BC, organic, and sulfate aerosol burdens (e.g., AMAP, 2010; Eckhardt et al., 2013; Breider et al., 2014; Ferrero et al., 2016; Gunsch et al., 2017; Creamean et al., 2018; Kirpes et al., 2018). The contribution of high-latitude flaring emissions to Arctic BC concentration is controversially discussed. While some studies suggest gas flaring to be an important source of BC, particularly in winter and spring (Stohl et al., 2013; Xu et al., 2017; Leaitch et al., 2018; Zhu et al., 2020), others provide evidence that flaring plays a minor role (Winiger et al., 2017, 2019).

Although considerable advances have been achieved in recent years, the majority of our current understanding of aerosol properties in the summertime Arctic is obtained from ground-based and shipborne measurements. In particular, airborne studies that attribute aerosol physical and chemical properties to sources are sparse, especially in summer (Radke and Hobbs, 1989; Brock et al., 1989; Paris et al., 2009; Schmale et al., 2011; Quennehen et al., 2011; Matsui et al., 2011a, b; Kupiszewski et al., 2013; Creamean et al., 2018). However, detailed knowledge on the vertical structure of aerosol composition and mixing state as well as the related particle sources, formation, and aging processes are necessary to have a predictive understanding of our Arctic climate system; yet, the vertical distribution of summertime Arctic aerosol is not well represented in Arctic models (e.g., Quinn et al., 2008; Moore et al., 2011; Eckhardt et al., 2015; Sato et al., 2016; Sand et al., 2017; Willis et al., 2018; Abbatt et al., 2019; Schmale et al., 2021). Our study thus focuses on processes and sources controlling summer Arctic aerosol, using vertically resolved measurements of aerosol properties and trace gases, together with Lagrangian air mass history analysis. To our knowledge, this is the first comprehensive source attribution study of summertime Arctic aerosol composition, combining airborne single particle and bulk chemical composition methods, with focus on the vertical structure.

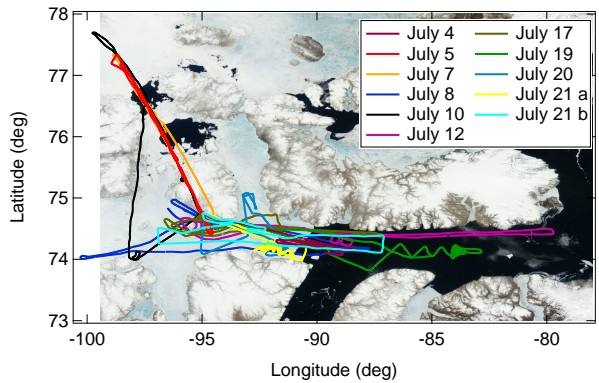

**Figure 1.** Satellite image (visible range from MODIS) from 4 July 2014 with a compilation of flight tracks conducted during 4 - 21 July 2014 (indicated with different colors). On 21 July 2014, two flights were performed indicated by a and b in the legend.

## 2 Experimental and Modeling Methods

### 2.1 Airborne Arctic field experiment NETCARE 2014

Motivated by limited knowledge of summertime Arctic aerosol processes, the aircraft campaign NETCARE 2014 took place from Resolute Bay, Nunavut (Canada) from 4 - 21 July 2014 (Fig. 1). This was part of the NETCARE project (Network on Climate and Aerosols: Addressing Key Uncertainties in Remote Canadian Environments; Abbatt et al., 2019). Resolute Bay faces the Northwest Passage with Lancaster Sound near Baffin Bay and Nares Strait (see Fig. 1 in Köllner et al., 2017). Airborne measurements of aerosol physical and chemical properties, trace gases as well as meteorological parameters were performed onboard the AWI Polar 6 aircraft, a DC-3 aircraft modified to a Basler BT-67 for operation in cold and harsh environments (Herber et al., 2008). Eleven flights were conducted from Resolute Bay (Fig. 1). The total sampling time was about 44 hours, of which roughly half of the time has been spent each in the BL and the FT. Flight tracks covered altitudes typically up to 3.5 km, while the flight on 8 July reached altitudes up to 6 km.

The measurements took place largely over remote areas characterized by open water (Lancaster Sound and polynyas north of Resolute Bay), sea ice, and Arctic vegetation (Fig. 1). The sharp transition between open water and sea ice was situated approximately 150 km south-east of Resolute Bay (Fig. 1). Local anthropogenic pollution might have affected the measurements, but is primarily related to the sparse Arctic settlements with domestic activities, traffic, landfill and waste burning as well as local airports, and electric power generators (Aliabadi et al., 2015). The nearest neighboring communities are Arctic Bay and Grise Fjord approximately 360 km from Resolute Bay. In summary, the remoteness of the Resolute Bay region provided a well-suited location for studying the pristine summertime Arctic atmosphere with access to open water regions.

## 2.2 Aerosol particle composition measurements

### 2.2.1 Single particle mass spectrometer - ALABAMA

The Aircraft-based Laser ABlation Aerosol MAss spectrometer (ALABAMA) was deployed on the Polar 6 aircraft during the Arctic field experiment, to provide on-line measurements of single particle chemical composition and size. The instrument has been described before in detail (Brands et al., 2011; Roth et al., 2016; Schmidt et al., 2017; Köllner et al., 2017) and is only briefly reviewed here. Particles enter the system through a critical orifice (for details see Molleker et al., 2020). The following Liu-type aerodynamic lens focuses particles into a narrow beam (Liu et al., 1995a, b, 2007). Further, particles are optically detected by scattering light when passing through two orthogonal detection laser beams ($\lambda = 405$ nm). The scattered light is focused by elliptical mirrors and further detected by photomultiplier tubes. This setup provides the particle's time of flight and thus its velocity. The vacuum aerodynamic diameter ($d_{\mathrm{va}}$) can be derived from the velocity, using a calibration with particles of known size, density, and shape (here, monodisperse PSL particles in a size range between 190 nm and 800 nm). During NETCARE 2014, the ALABAMA analyzed particles in a size range between 320 nm and 870 nm ($d_{\mathrm{va}}$). The lower and upper size cut-offs are defined by the size range that include 98 % of all particles analyzed during NETCARE 2014. The successfully detected and sized particles are ablated and ionized in the high-vacuum region by a single triggered laser shot ($\lambda = 266$ nm). The generated ions are guided into a Z-shaped Time-of-Flight mass spectrometer that provides bipolar mass spectra of individual particles. Details on the data analysis can be found in Sect. 2.4

### 2.2.2 Aerodyne High-Resolution Time-of-Flight aerosol mass spectrometer - HR-ToF-AMS

Sub-micron ensemble aerosol composition in a size range between 75 and 900 nm was measured by an Aerodyne High-Resolution Time-of-Flight aerosol mass spectrometer (HR-ToF-AMS), complementary to the ALABAMA single particle analysis. The HR-ToF-AMS is based on particle thermal vaporization by contact with a heated surface ($\sim 650\,^{\circ}$C) followed by electron impact ionization (e.g., DeCarlo et al., 2006), in contrast to the ALABAMA laser-induced ablation/ionization process. The HR-ToF-AMS allows quantitative mass concentration measurements of non-refractory aerosol particle components. In practice, organic matter, nitrate, sulfate, ammonium, and methanesulfonic acid (MSA) are detected. To note, we further present the ratio of organic-to-sulfate mass concentrations measured by the HR-ToF-AMS. For further details see Willis et al. (2016, 2017) and Table S3. Both aerosol mass spectrometers, the ALABAMA and the HR-ToF-AMS, were not operated at altitudes above 3.5 km to avoid damages under low pressure conditions in the cabin.

### 2.2.3 Single Particle Soot Photometer - SP2

Concentration of particles containing refractory black carbon (rBC) was acquired with a Single Particle Soot Photometer (SP2) manufactured by DMT (Schwarz et al., 2006; Gao et al., 2007). The instrument includes a continuous intra-cavity Nd:YAG laser ($\lambda = 1064$ nm) to classify individual particles as incandescent. The peak incandescence signal is linearly related to rBC

mass. The instrument detects particles in a size range between 85 and 700 nm. For further details see Schulz et al. (2019) and Table S3.

## 2.3 Complementary experimental methods

Meteorological and state parameters were measured by the AIMMS-20 by Aventech Research Inc., integrated on the Polar 6 aircraft. The module Air-Data Probe provides temperature, pressure, relative humidity, wind direction, wind speed, and the three-dimensional aircraft-relative flow vector (true air speed, angle of attack, and side slip). The second module, a GPS, processes satellite navigation signals and determines aircraft three-dimensional position and altitude (Aventech, 2018).

Carbon monoxide (CO) mixing ratios were measured by an Aero-Laser ultra-fast CO monitor (model AL 5002), based on the fluorescence of CO in the vacuum ultraviolet (VUV) at 150 nm (Scharffe et al., 2012; Aero-Laser GmbH, 2013; Wandel, 2015; Bozem et al., 2019). Here, CO is used as an indicator of air masses influenced by combustion sources, including fossil fuel, and biomass burning combustion (Andreae and Merlet, 2001). Further details can be found in Bozem et al. (2019) and Table S3.

Number concentrations of particles between 85 nm and 1000 nm were measured with the Ultra-High Sensitivity Aerosol Spectrometer (UHSAS), manufactured by DMT. The instrument is based on particle light scattering technique. Particles that cross the beam of a laser ($\lambda = 1054$ nm) are counted and sized by their scattering signals. The intensity of the scattered light provides information on the optical diameter ($d_{opt}$) (e.g., Cai et al., 2008; Schulz et al., 2019). For further details see Schulz et al. (2019) and Table S3. For the conversion of the ALABAMA fraction into number concentration (see Sect. 2.4), UHSAS number concentrations in a size range between 320 nm and 870 nm ($N_{>320}$) are used (see Sect. 2.2.1). Sampling strategy and inlet setups are discussed in detail by the following related publications: Leaitch et al. (2016); Willis et al. (2016, 2017); Aliabadi et al. (2016b); Burkart et al. (2017); Köllner et al. (2017); Bozem et al. (2019); Schulz et al. (2019).

## 2.4 Analysis of the ALABAMA single particle mass spectrometer data

The marker ion method is used to classify particles measured by the ALABAMA (see Köllner et al., 2017). The ion marker method classifies particles based on the presence of pre-selected species that are of interest. This approach is not dependent on the absolute value of the ion peak areas within one spectrum, but only on the presence of a certain ion peak area above a defined threshold value. In principle, four analysis steps are required. First, a pre-selection of chemical compounds that are of interest for the study is necessary. Second, ion markers of laboratory-generated particles with known composition are compared to ions from ambient particles detected during the field experiment. Third, a threshold for the ion peak area is determined to distinguish the ion signal from background noise (for details see Köllner et al., 2017). Finally, internal and external mixtures of chemical compounds are distinguished by checking if the respective ions occur in the same particle mass spectrum or in separate particle mass spectra. Both types of mixtures can be presented in a decision tree. The decision tree is structured as follows: upper/lower branches refer to positive/negative response for whether ion markers of the respective substance are above/below the ion peak threshold. This study focuses on six particle types based on the presence of the following substances (see Table 1): sodium and/or chloride and/or nitrate (Na/Cl/Nitrate), elemental carbon (EC), nitrate ($NO_3$), trimethylamine (TMA), dicar-

**Table 1.** Particle classification by marker species and associated ion markers applied in this study. References from laboratory and field studies using single particle mass spectrometry are indicated by numbers that are defined below. This table is partly adapted from Köllner et al. (2017).

| Marker species (abbreviation) | Ion markers | References (lab/field) | Comments |
|---|---|---|---|
| Trimethylamine (TMA) | $m/z$ +59 $((CH_3)_3N^+)$ and +58 $(C_3H_8N^+)$ | 1,2/3,4,5 | |
| Sodium and/or chloride and/or nitrate (Na / Cl / Nitrate) | $m/z$ +23 $(Na^+)$ and (at least two of the following ions) +46 $(Na_2^+)$, +62 $(Na_2O^+)$, +63 $(Na_2OH^+)$; (at least one of the following isotopic patterns/ions) +81/83 $(Na_2Cl^+)$, -35/37 $(Cl^-)$, -93/95 $(NaCl_2^-)$, -46 $(NO_2^-)$, -62 $(NO_3^-)$ | 6,7,8/9,3 | Isobaric interference with MSA at $m/z$ -95 |
| Elemental carbon (EC) | (at least six of the following ions) $m/z$ +36, +48, ..., +144 $(C_{3-12}^+)$ and/or (at least six of the following ions) $m/z$ -36, -48, ..., -144 $(C_{3-12}^-)$ | 7,8/3,10 | Except $m/z$ -96 due to isobaric interference with $SO_4^-$ |
| Low-molecular weight dicarboxylic acids (DCA) | (at least one of the following ions) $m/z$ -89 $(C_2HO_4^-)$, $m/z$ -103 $(C_3H_3O_4^-)$, $m/z$ -117 $(C_4H_5O_4^-)$ | 11/12,13,14 | representative: oxalic acid malonic acid succinic acid |
| Nitrate $(NO_3)$ | (at least one of the following ions) $m/z$ -46 $(NO_2^-)$, -62 $(NO_3^-)$ | 10,11/3,7 | |
| Potassium (K) | $m/z$ +39/41 $(K^+)$ | 7, Fig. S6 (this study)/3,15,16 | |

Given reference numbers are defined as follows: [1]Angelino et al. (2001),[2]Köllner et al. (2017),[3]Roth et al. (2016),[4]Healy et al. (2015),[5]Rehbein et al. (2011),[6]Prather et al. (2013),[7]Schmidt et al. (2017),[8]Brands (2009),[9]Sierau et al. (2014),[10]Brands et al. (2011),[11]Silva and Prather (2000),[12]Lee et al. (2003),[13]Sullivan and Prather (2007),[14]Yang et al. (2009),[15]Hudson et al. (2004),[16]Pratt et al. (2011).

boxylic acids (DCA), and potassium (K). We choose these substances because they are well suited to study aerosol sources as well as aerosol processing along transport that are subjects of this work, such as marine, vegetation fires, and anthropogenic sources. We further analyzed ion peaks that might correspond to levoglucosan, but the marker ions are ambiguous (see Sup-

plement Sect. 4). As a result, we will not discuss levoglucosan in this analysis.

In the following, we present the conversion of unscaled ALABAMA measurements into quantitative particle number concentrations. Several studies using single particle mass spectrometry applied a similar scaling procedure (e.g., Qin et al., 2006; Healy et al., 2013; Gunsch et al., 2018; Froyd et al., 2019). Based on particle classification, the fraction of the ALABAMA particle types is calculated (Fig. S1a). To determine particle fractions, total analyzed (i.e., detected+ablated/ionized) particles and specific particle types that are of interest are categorized into bins of a simultaneously measured variable (see horizontal

axis in Fig. S1). For each bin, the particle fraction is defined as the number of particles of the specific type divided by the number of all analyzed particles ($PF$; see Fig. S1a). To note, the following use of the word *fraction* always refers to the number fraction measured by the ALABAMA. Bin widths are chosen to guarantee an acceptable level of both statistical significance and resolution. Bins containing less than 20 spectra in total were excluded from the analysis. Further, the particle fraction is scaled to number concentrations, using measurements of a co-located quantitative reference instrument, here the UHSAS

onboard the Polar 6 (see Sect. 2.3). The UHSAS number concentration in a size range between 320 nm and 870 nm ($N_{>320}$; Fig. S1b) is used. The selected UHSAS lower and upper size cut offs refer to the ALABAMA size range measured during NETCARE 2014 (see Sect. 2.2). Conceptually, scaled number concentrations of the ALABAMA particle types are determined by multiplying the particle fraction with the averaged UHSAS number concentration in each bin ($PF \cdot N_{>320}$; Fig. S1c). An example of this scaling procedure is given in the Supplement Sect. 1.

By using this scaling method, it is assumed that the $d_{opt}$ measured by the UHSAS and the $d_{va}$ measured by the ALABAMA are equal. It is further assumed that the ALABAMA transmits, detects, and ablates/ionizes all particles with the same efficiency. These assumptions introduce uncertainties in the scaled number concentrations, as the detection efficiency (DE) and the hit rate are observed to be dependent on several factors. The DE is size-dependent, mainly caused by the transmission efficiency of the aerodynamic lens and the optical detection (Zhang et al., 2004; Liu et al., 2007; Brands et al., 2011). However, a size-

dependent scaling factor was not deployed in this study, owing to the low particle counting statistics per bin. As a result, the scaling method may lead to uncertainties when determining the concentration of particles smaller than approximately 400 nm. Likewise, the hit rate in laser ablation/ionization technique is dependent on various particle characteristics, such as size, composition, and shape (Thomson et al., 1997; Kane et al., 2001; Moffet and Prather, 2009; Brands et al., 2011). The hit rate can thus vary during ambient measurements, which may cause a problem when interpreting the fraction of particle types. Further

details on the hit rate variability during NETCARE 2014 are given in the Supplement Sect. 1.

Given that the scaled number concentration is estimated under several assumptions, we solely discuss trends in number concentrations in comparison with other parameters to identify sources of different particle types, rather than discussing the absolute number concentration. In addition, we will only discuss trends that are clearly visible in both the number fraction and scaled concentration. For clarity, we focus on the scaled number concentration ($PF \cdot N_{>320}$) in the following sections. Additional

235 figures showing particle fraction and averaged UHSAS number concentration are in given in the Supplement Sect. 6.

## 2.5 Air mass history modeling

### 2.5.1 FLEXPART

Air mass history analysis is crucial for understanding aerosol sources and transport processes, both of which control aerosol properties in Arctic regions. In this study, we use the Lagrangian FLEXible PARTicle dispersion model (FLEXPART, version 10.0). FLEXPART is a comprehensive tool for modeling atmospheric transport (e.g., Stohl, 1998; Stohl et al., 2003, 2005) and is widely used in Arctic studies (e.g., Stohl, 2006; Paris et al., 2009; Schmale et al., 2011; Evangeliou et al., 2019; Zhu et al., 2020). In this study, FLEXPART was operated backward in time with operational data from the ECMWF with 0.25° horizontal and 3 hours temporal resolution as well as with 137 vertical hybrid sigma-pressure levels. Tracer air parcels (usually called tracer particles) are released from a certain receptor location and followed for 15 days backward in time (Stohl et al., 2003; Seibert and Frank, 2004; Stohl et al., 2005). The receptor location depends on airborne sampling location. In detail, 20000 tracer particles were initialized each 10-min sampling interval along the flight track. By this approach, between 16 and 29 releases were created for each flight, depending on flight duration. The release volume of the tracer particles for each 10-min interval was set by the maximum and minimum geographic locations during this time segment. The main variable calculated by FLEXPART is the so-called Potential Emission Sensitivity (PES) that describes the sensitivity of air sampled at the receptor to source emission locations (Seibert and Frank, 2004; Stohl et al., 2005; Hirdman et al., 2010). The PES value has the dimension of seconds and essentially is proportional to the parcel's residence time in a particular output grid cell (Seibert and Frank, 2004; Hirdman et al., 2010). It is assumed that each source contributes with unit strength in the respective cell. Further, the transported tracer is inert, which means no impact of precipitation and no chemical reactions are considered in FLEXPART (Stohl et al., 2003; Hirdman et al., 2010). The PES function is provided every six hours (i.e., 60 values for a 15 day backtrajectory), with uniform horizontal grid spacing of 0.25°, and with five vertical levels (400, 1000, 2000, 5000, and 15000 m).

FLEXPART output combined with geographical locations of emission sources may provide a more detailed analysis of the relationship between Arctic aerosol composition and sources. This analysis requires the following data processing steps: First, the total residence time of air masses in each grid cell is obtained by integrating PES data in time for a given vertical layer. It is assumed that surface emissions into this vertical layer, the so-called footprint layer, are mixed instantaneously by turbulence in the BL. Therefore, the lowest vertical level (0 - 400 m) is applied for most ground sources (e.g., Stohl et al., 2013), except for vegetation fires related to buoyant effects (see details in the Supplement Sect. 2.1). Second, we show maps that specify geolocations of source sectors and source regions (see Sect. 2.5.2). Third, the prior obtained two-dimensional (latitude and longitude) PES map is folded with the two-dimensional map of source regions and sectors. Finally, each grid cell is summed to obtain the total residence time of sampled air mass spent over a specific source within the 15 days prior to sampling and within the vertical footprint layer. In general for this study, the PES fraction is defined as the ratio of the PES of sampled air above source regions/sectors within the footprint layer of the model domain over a 15-days integrated backward simulation compared to the PES of sampled air in the total vertical column of the model domain over a 15-days integrated backward simulation.

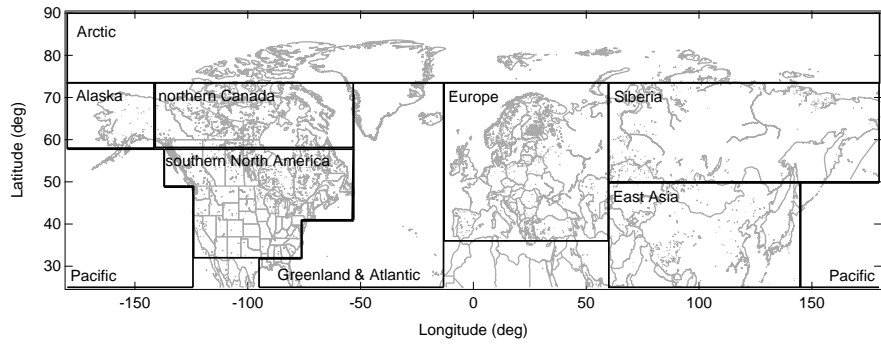

**Figure 2.** Map showing selected source regions: Arctic, Alaska, northern Canada, southern North America, Pacific, Greenland & Atlantic, Europe, Siberia, and East Asia.

**Table 2.** Geographical extent of chosen source regions

| Region | Latitude (deg) | Longitude (deg) | Latitude (deg) | Longitude (deg) | Latitude (deg) | Longitude (deg) |
|---|---|---|---|---|---|---|
| Arctic (July 2014*) | +73.5 - +90 | −180 - +180 | | | | |
| Europe | +36 - +73.5 | −13 - +60 | | | | |
| Siberia | +50 - +73.5 | +60 - +180 | | | | |
| East Asia | +25 - +50 | +60 - +145 | | | | |
| Alaska | +58 - +73.5 | −141 - −180 | | | | |
| northern Canada | +58 - +73.5 | −53 - −141 | | | | |
| southern N. America | +32 - +58 | −76 - −124 | +49 - +58 | −124 - −137 | +41 - +58 | −53 - −76 |
| Pacific | +25 - +50 | +145 - +180 | +25 - +49 | −124 - −180 | +49 - +58 | +137 - +180 |
| Greenland & Atlantic | +25 - +73.5 | −13 - −53 | +32 - +41 | −53 - −76 | +25 - +32 | −53 - −95 |

*according to Bozem et al. (2019)

### 2.5.2 Source regions and sectors

Geolocations of selected source regions (e.g., Europe) and sectors (e.g., vegetation fires) are visualized using two-dimensional
maps with horizontal grid spacing of 0.25°. Geographical regions that likely contribute to Arctic aerosol in summer comprise
primarily northern Eurasia, North America, East Asia, and the Arctic itself (e.g., Stohl, 2006; Shindell et al., 2008; Bourgeois
and Bey, 2011; Liu et al., 2015; Yang et al., 2018). However, we consider Europe and Siberia as well as Alaska, northern
Canada (nCa), and southern North America (sNA) separately due to differences in vegetation fire occurrence, industrial sources,
flaring spots, population density (Figs. 3a-d, respectively), and meteorological conditions. Large marine areas of the Pacific
and Atlantic Oceans are also included due to their vicinity to Arctic regions. Definitions of the source regions were chosen
according to prior Arctic studies (Stohl, 2006; Shindell et al., 2008; Liu et al., 2015; Yang et al., 2018), except for the Arctic.
For our July 2014 study, Arctic regions are specified as located north of 73.5° N, based on the July 2014 polar dome location

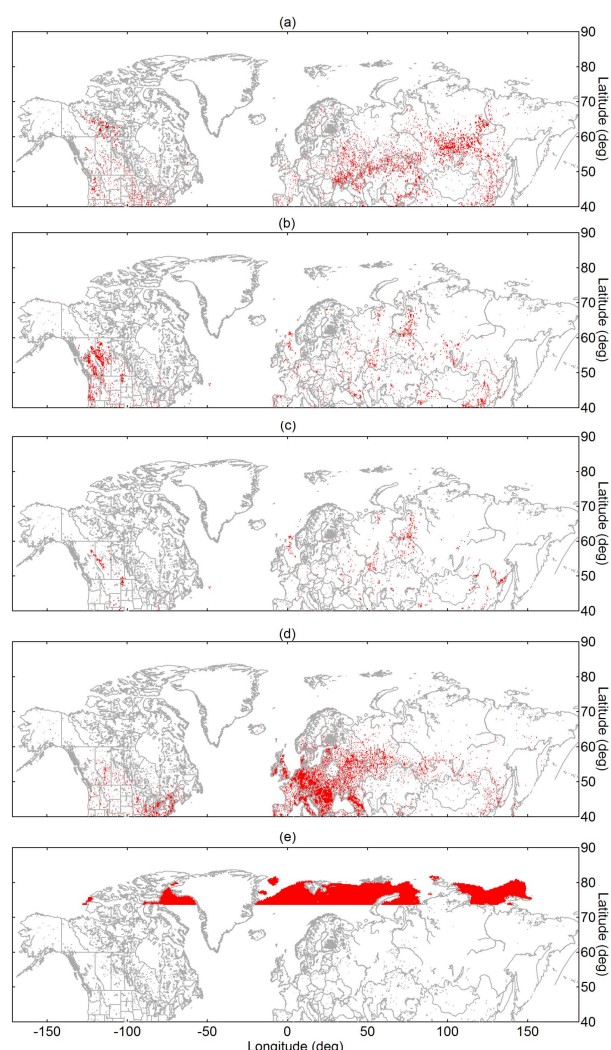

**Figure 3.** Map showing the geolocation of selected source sectors: (a) vegetation fires, (b) industry (including flaring), (c) flaring, (d) population, and (e) Arctic open water. Panel (a) shows accumulated fires from 19 June to 21 July 2014. Red pixels indicate the presence of the respective source sector irrespective of the particular emission strength.

determined by Bozem et al. (2019). Table 2 and Figure 2 provide the geographical extent of the chosen source regions.

Details on creating maps (Fig. 3) that specify the geolocation of the following source sectors: vegetation fires, industry, flaring, population, and Arctic open water, are given in the Supplement Sect. 2. Briefly, for identifying vegetation fire spots, we used data from the Visible Infrared Imaging Radiometer Suite (VIIRS) onboard the Suomi National Polar-Orbiting Partnership spacecraft. These measurements provide sub-pixel thermal anomalies by signals in mid- and thermal infrared spectral range (Schroeder et al., 2014; Schroeder and Giglio, 2018). However, thermal anomalies typically include both active vegetation fires and industrial heat sources by fossil-fuel combustion etc. Therefore, anthropogenic heat sources, such as industry, flaring, and

densely populated regions, are separately mapped (as explained below) and subsequently excluded from the thermal anomaly map. Residual heat sources in Fig. 3a thus present vegetation fires. Industrial areas, including flaring, are obtained by using quantitative estimates of infrared source emission temperatures from the Nightfire product from VIIRS (Figs. 3b and c). This method is taking advantage of absent solar reflectance in short-wave infrared spectral range during night (Elvidge et al., 2013, 2016). Emission temperatures typically exhibit bimodal distributions by a combination of gas flares dominating the upper mode (peaking in the 1600 - 2000 K range) and other industrial heat sources as well as vegetation fires dominating the lower mode (peaking in the 800 - 1200 K range; Elvidge et al., 2016; Liu et al., 2018). Geolocation data on population are provided online (MaxMind, 2018). The criteria for the map grid cell to be classified as populated is given by more than 1000 people living in a 0.25° x 0.25° area (Fig. 3d). Finally, we used measurements from the Special Sensor Microwave Imager/Sounder (SSMIS) onboard the Defense Meteorological Satellite Program (DMSP) to assess Arctic sea ice coverage and open water areas in July 2014 (Fig. 3e). Data on sea ice coverage are based on detected brightness temperature (Cavalieri et al., 1996, updated yearly). In general, these approaches classify whether the source sector is present or absent in the grid cell irrespective from source strengths, rather than providing quantitative source-specified emission data.

Different source sectors can contribute to air mass history within the 15 days prior to sampling. The Supplement Sect. 9 shows comparisons of PES fractions between different source sectors. It was found that air masses with high Arctic open water PES fraction were largely isolated from other sources; whereas anthropogenic sources (industrial and populated areas) can contribute on either side to air mass history by their close proximity. However, a pre-processing of the PES fractions was applied, if possible, in order to differentiate different source sector contributions from each other. For example, we could separate contributions of vegetation fires from anthropogenic sources. Further details can be found in the Supplement Sect. 9.

## 3 Results and discussions

### 3.1 Single particle chemical composition

By means of the ALABAMA, 10137 particles were analyzed during the NETCARE 2014 study when sampling outside clouds; 94 % of the spectra include size information; 78 % of the spectra have dual polarity (Fig. 4). Potential reasons for the lack of negative ions are discussed in Köllner et al. (2017). It might be that single-polarity spectra are preferably produced in high relative humidity environments (Neubauer et al., 1998; Spencer et al., 2008), in particular in marine environments (Guasco et al., 2014). Particle classification by the ion marker method introduced in Sect. 2.4 is illustrated here by means of a decision tree in Fig. 4. Particle groups that include less than 300 particles ($\sim 3\%$) are not further sub-classified due to the low counting statistics. This means EC- and Na/Cl/Nitrate-containing particles are excluded from other types due to their low number. In contrast, particles that contain substances such as nitrate, TMA, and/or DCA and that are not internally mixed with EC and Na/Cl can be assigned to several types. TMA-, DCA-, and sub-K-containing particles are the most prominent types with relative contributions of 23 %, 31 %, and 22 % to the total number of analyzed particles, respectively (indicated by yellow, brown and green filling in Fig. 4). In contrast, particles containing Na/Cl/Nitrate, EC, and nitrate (externally mixed with Na and Cl) account for 2 %, 2 %, and 7 %, respectively. 24 % of all mass spectra are not considered for further analysis (gray filling

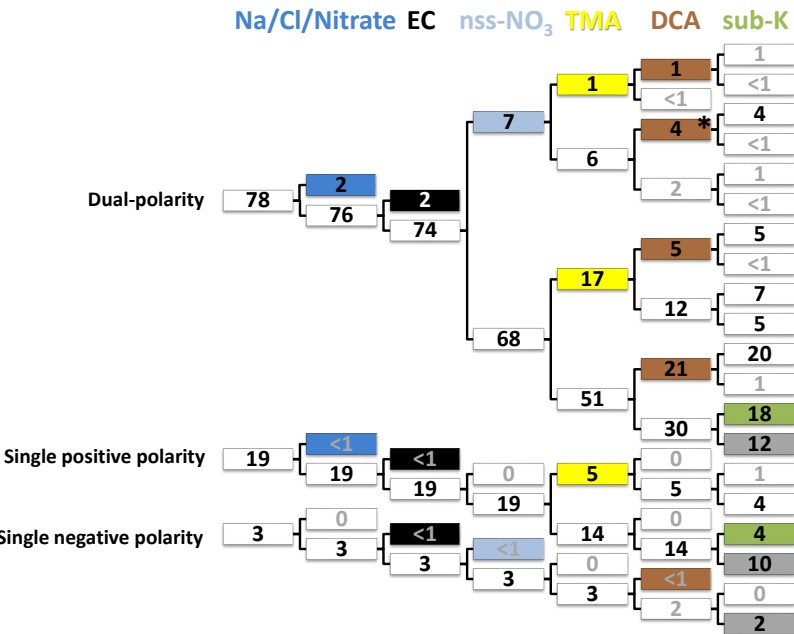

**Figure 4.** Classification of 10137 single particle spectra based on the ion marker method and illustrated with the decision tree (see Sect. 2.4). Upper/lower branches refer to the presence/absence of substances in the particle mass spectra. Numbers are given as relative abundances in % normalized to 10137 particles. The initial classification provides three groups of particle mass spectra with dual-, single positive, and single negative polarity. Particle types, indicated by colors, contain: Na/Cl/Nitrate (dark blue), EC (black), non-sea-spray-(nss)-NO₃ (light blue), trimethylamine (TMA) (yellow), dicarboxylic acids (DCA) (brown), potassium (sub-K) (green), and "Remaining" (gray filling). As an example, 4 % of 10137 particles (indicated by the asterisk) contain DCA and nss-nitrate (from right to left). The remaining particles are further sub-classified in the Supplement Fig. S9

in Fig. 4), however, those remaining mass spectra are further sub-classified with marker species sulfate, ammonium, MSA etc. (see Fig. S9). In the following, mean spectra of the main particle types (Fig. 5) combined with additional ion signals (Table S2) provide an overview of the averaged chemical composition. To note, percentages given in this study always refer to number percentages measured by the ALABAMA.

The term particulate DCA implies the presence of oxalic, malonic, and/or succinic acid (see Table 1), with oxalic acid as most abundant (in 86 % of DCA particles), followed by succinic acid with 41 % of DCA particles and malonic acid with 38 % of all DCA particles (not shown). It should be noted that the ALABAMA detects oxalate-, malonate-, and succinate ions that most likely originate from oxalic, malonic, and succinic acid, respectively (Silva and Prather, 2000). The majority of the DCA spectra show the concurrent presence of oxidized organics at m/z -45, -59, -71, and -73 (Fig. 5e and Table S2). From our ALABAMA analysis, we can infer that particulate DCA was partly internally mixed with particles containing sodium, chloride, and/or nitrate (Na/Cl/Nitrate), which are interpreted as sea spray particles (Table S2 and Fig. S8). In detail, the presence of DCA was found in 60 % of those sea spray particles. The presence of DCA together with sea spray had been observed earlier

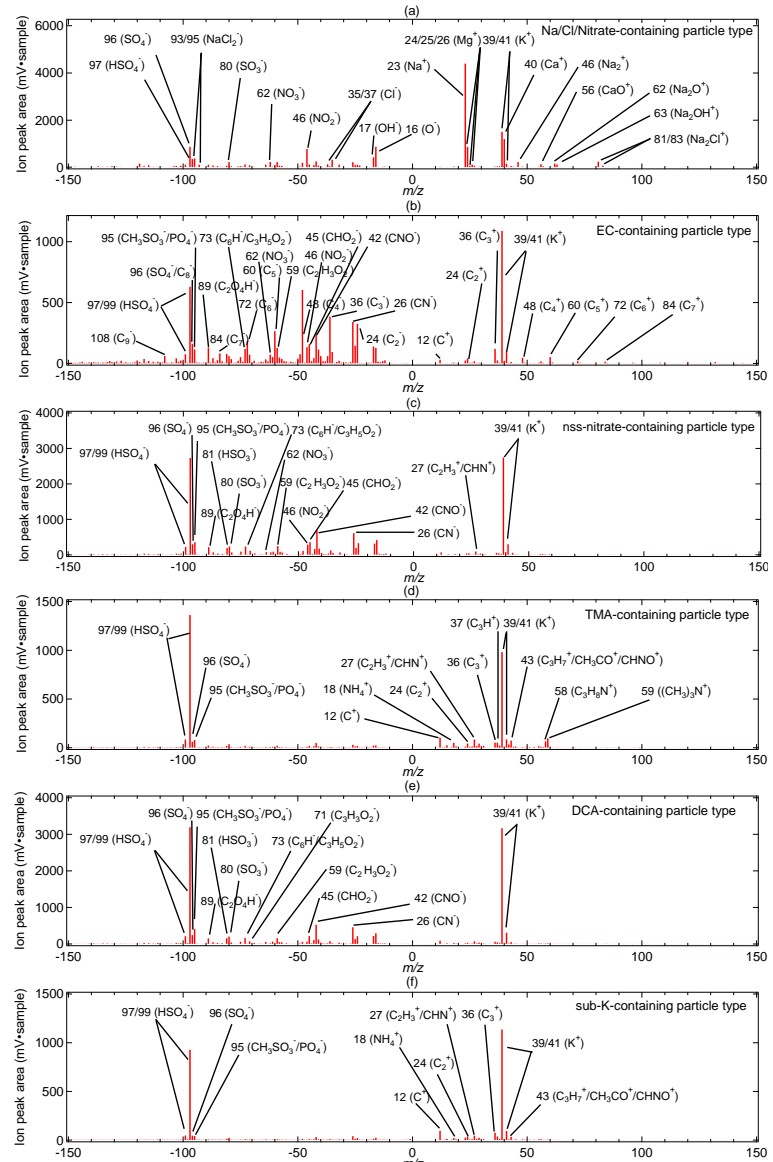

**Figure 5.** Bipolar mean spectra of the ALABAMA particle types: (a) Na/Cl/Nitrate-containing (195 particles out of 10137 $\widehat{=}$ 2 %), (b) EC-containing (188 particles out of 10137 $\widehat{=}$ 2 %), (c) nss-nitrate-containing (688 particles out of 10137 $\widehat{=}$ 7 %), (d) TMA-containing (2325 particles out of 10137 $\widehat{=}$ 23 %), (e) DCA-containing (3150 particles out of 10137 $\widehat{=}$ 31 %), and (f) sub-K-containing (2210 particles out of 10137 $\widehat{=}$ 22 %) .

during measurement campaigns with a focus on marine air (Sullivan and Prather, 2007), also in Arctic regions (Kerminen et al., 1999). However, the majority of DCA were not internally mixed with sea spray (Figs. 4 and 5e). Moreover, 14 % of those DCA-containing particles (without sea spray) were internally mixed with TMA and MSA (Figs. 4 and 5e). In our prior study,

we concluded that particulate TMA in the Arctic summer results from secondary formation, owing to the external mixture of TMA and primary components, such as sodium (Köllner et al., 2017). Thus, together with the secondary nature of TMA and MSA (Hoffmann et al., 2016), the internal mixture of DCA with TMA and MSA suggests that DCA results from the secondary (potentially aqueous) conversion of precursor gases (e.g., Kawamura and Bikkina, 2016).

Nitrate was also externally mixed with sea spray particles, hereafter referred to as non-sea-spray (nss)-nitrate. The averaged chemical composition of EC- and nss-nitrate-containing types provide evidence for internal mixing with secondary substances such as sulfate, MSA, and oxygen-containing organic matter (OM; Figs. 5b-c and Table S2). In addition, 20 % of the EC particles were internally mixed with nitrate (Fig. 5b and Table S2). More than 50 % of the nss-nitrate- and EC-containing particle spectra include ion signals of DCA (Fig. 4), particularly oxalic acid at m/z -89 (Fig. 5b-c). The majority of the nss-nitrate particles (85 %) were internally mixed with potassium (Figs. 4 and 5c). In contrast to that, the sub-K-containing particle type presents a sub-group that includes potassium-containing particles that are externally mixed with sea spray, EC, nitrate, TMA, and DCA (Figs. 4 and 5f). The mean spectrum is dominated by the presence of potassium and sulfate as well as some small organic peaks in the cation spectrum (Fig. 5f and Table S2). First conclusions on aging and formation of the different particle types were drawn based on the averaged chemical composition and mixing. The following sections provide a detailed discussion on aerosol sources and formation processes during two distinct meteorological periods.

## 3.2 Arctic air mass period

Meteorological conditions combined with air mass history play a major role for understanding Arctic aerosol composition influenced by local as well as distant sources and transport. The synoptic situations during NETCARE 2014 have been discussed in previous publications (see Bozem et al., 2019; Burkart et al., 2017; Köllner et al., 2017; Leaitch et al., 2016). Air mass history derived by FLEXPART is summarized in Figs. 6 and 14 as a function of sampling altitude. The synoptic conditions changed over the course of NETCARE 2014 from an initial Arctic air mass period (July 4-12, $\sim$ 26 measurement hours) to a southern air mass period (July 17-21, $\sim$ 19 measurement hours) with a transition in between (Burkart et al., 2017; Bozem et al., 2019). Since the different synoptic conditions have a significant impact on the distributions of the observed quantities, we analyze the measurements for each period individually.

### 3.2.1 Meteorological overview and air mass history

During the Arctic air mass period, the measurement region was under the influence of a prevailing high-pressure system with generally clear skies, some low-level stratocumulus clouds (Leaitch et al., 2016), low wind speeds, and a stable stratified shallow BL (Bozem et al., 2019). Inferred from the vertical profile of pseudoequivalent potential temperature, the local upper BL height was on average between 250 m and 450 m, including a surface-based temperature inversion (Köllner et al., 2017). The representativeness of the Arctic air mass period in July 2014 with respect to climatological conditions is discussed in the Supplement Sect. 3. Briefly, synoptic situations were generally consistent with the climatology.

Air mass history during the first period reflects the concept of isentropic transport to Arctic regions during summer. Klonecki et al. (2003) and Stohl (2006) provide comprehensive analyses on the role of transport pathways into the Arctic troposphere

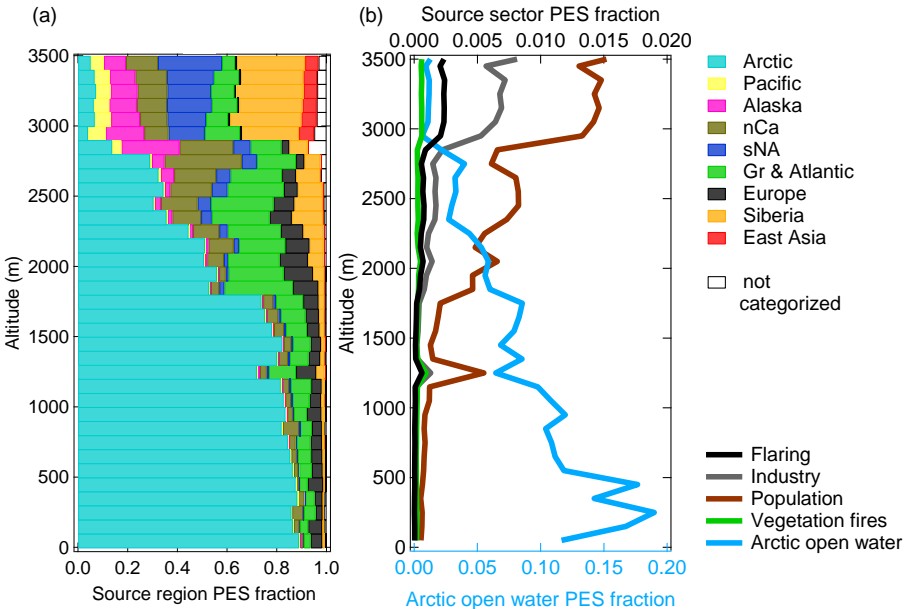

**Figure 6.** Vertically resolved cumulative contribution of (a) source regions and (b) source sectors to sampling altitude during the Arctic air mass period (July 4-12). For details on flight tracks see Fig. 1. The FLEXPART derived contribution is expressed as a fraction of the potential emission sensitivity (PES) in the model domain lowest vertical level (0 - 400 m, except for vegetation fires) over a 15-days backward simulation. A vertical injection layer between 1 and 5 km is applied for vegetation fires (see details in Sect. 2.5.1). Acronyms in (a) are defined as follows: nCa = northern Canada; sNA = southern North America; Gr = Greenland. The white parts depict areas that are not categorized in the selected geographical regions. Bottom and top axes in (b) represent PES fractions of Arctic open water and other source sectors, respectively.

across seasons. Both prior studies show that the Arctic summertime lower troposphere is quite isolated from southern latitude sources as diabatic low-level transport into the polar dome are largely absent. This is in line with our results. We found that the near-surface regions were largely isolated from the rest of the atmosphere, whereas regions aloft were episodically influenced by air originating from southern latitudes (Fig. 6a). The distinct change in air mass history at around 3 km altitude depicts the

370 transition to measurement regions located outside of the polar dome (Fig. 6a; Bozem et al., 2019). The period is referred to as Arctic air mass period since air mass history 15 days prior to sampling shows a dominant influence of air masses from the high Arctic. Inside the BL, we predominantly sampled air masses that had resided in the Arctic regions (up to 90 % in Fig. 6a). Particularly, a significant fraction of the high Arctic air masses has resided over open water (up to 20 % in Fig. 6b). Thus, air masses inside the BL originated to a large degree from pristine Arctic regions with mainly natural aerosol sources affecting

composition. Above the BL, influences of Arctic air masses were still pronounced (Fig. 6a). However, contributions from southern latitude regions, i.e. Europe, Siberia, northern Canada, Greenland, and the Atlantic Ocean, increased with altitude due to enhanced quasi-isentropic transport from these regions into the high Arctic (Fig. 6a). The abovementioned studies by Stohl (2006) and Klonecki et al. (2003) demonstrate that emissions in Siberia, Europa, and Asia can influence Arctic summertime

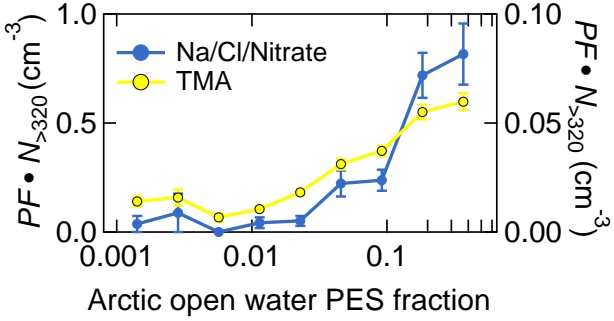

**Figure 7.** Aerosol composition measured by the ALABAMA as a function of FLEXPART derived PES fraction over Arctic open water areas during the Arctic air mass period. The panel represents the ALABAMA scaled number concentration ($PF \cdot N_{>320}$) with right axis referring to Na/Cl/Nitrate-containing (blue) and left axis referring to TMA-containing (yellow) particles. Uncertainty analyses are given in the Supplement Sect. 7.

composition in altitudes above the BL, which is consistent with our results.

An example of a transport from Europe towards the Arctic is shown in Figure S35. Air masses at low levels over Europe ascended along the upward sloping isentropes when they were transported poleward. Thus, the impact of the emissions from these source regions on our measurements becomes apparent at higher altitudes (Fig. 6a). Air mass history above 3 km altitude, above the polar dome (Bozem et al., 2019), indicates a sharp transition to a dominating influence of mid-latitude air masses (Fig. 6a). The contribution of Siberian regions to air mass history was highest in altitudes above 3 km. Consistent with several

modeling studies, contributions of regions south of 50-60° N (i.e., East Asia, Pacific, and southern North America) became significantly important only above 3 km, whereas sources in Europe and northern Canada frequently affected measurements in lower altitudes (e.g., Bourgeois and Bey, 2011; Sobhani et al., 2018; Yang et al., 2018). Thus, anthropogenic pollution in altitudes above 3 km can be attributed to sources in East Asia and southern North America (Fig. 6). In general during the Arctic air mass period, the contribution of vegetation fires was negligible (Fig. 6b). This finding is in line with low CO mixing ratios

(< 90 ppb$_v$) measured during the first period (Bozem et al., 2019).

### 3.2.2  Arctic marine influences on particle composition

The clean and pristine Arctic background conditions during the first period provide the unique opportunity to study the influence of Arctic marine sources on particle composition. This section complements the results presented in Köllner et al. (2017) and Willis et al. (2017), both analyzing particle chemical composition during the first period. It was shown that marine processes

dominated the presence of organic matter and MSA in low sulfate environments in the summertime Arctic BL (Willis et al., 2017; Köllner et al., 2017). The organic-to-sulfate and MSA-to-sulfate ratios peaked when the residence time over Arctic open water region was highest (Willis et al., 2017). As a new analysis compared to our prior studies, we present the aerosol composition measured by the ALABAMA as a function of FLEXPART derived PES fraction (see Sect. 2.5.1) over Arctic open

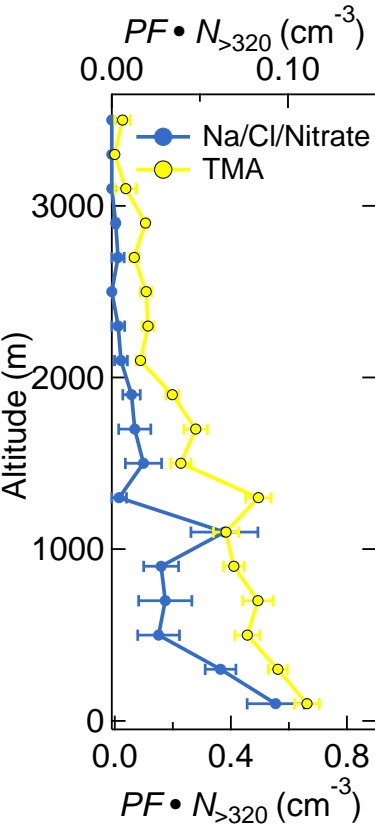

**Figure 8.** Vertically resolved aerosol composition measured by the ALABAMA during the Arctic air mass period. The panel represents the ALABAMA scaled number concentration ($PF \cdot N_{>320}$) with top axis referring to Na/Cl/Nitrate-containing (blue) and bottom axis referring to TMA-containing (yellow) particles. Uncertainty analyses are given in the Supplement Sect. 7.

water areas in Fig. 7 and the vertically resolved aerosol composition in Fig. 8. Particulate TMA was predominantly abundant when the air resided for more than 50 % of the 15 days (PES fraction) prior to sampling over Arctic open water areas (Fig. 7). Along with the marine influence, Na/Cl/Nitrate-containing particles were largely abundant when the residence time over Arctic open water regions was high (Fig. 7), linking those particles to locally emitted sea spray. In addition, both particle types are predominantly present in the Arctic marine BL (Fig. 8). Taken together, air mass history demonstrates that Arctic marine sources contributed to the abundance of particulate TMA internally mixed with MSA, sulfate, and/or other organics as well as sea spray particles in the summertime marine Arctic BL.

### 3.2.3 Long range transport influences on particle composition

In a next step, we will focus on the particle composition measured in the summertime Arctic lower FT. In contrast to particle composition in the Arctic marine BL, our analysis suggests that particle composition in the Arctic FT was influenced by long

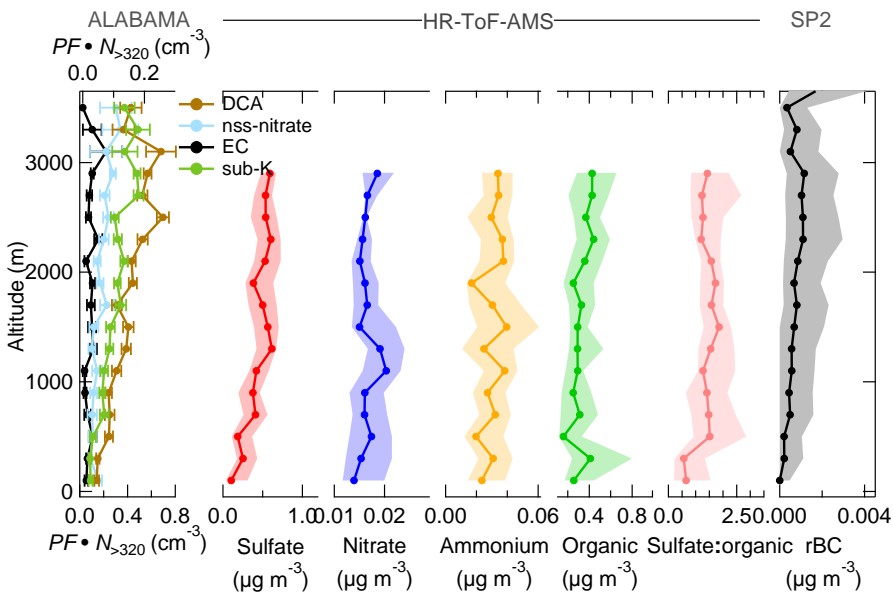

**Figure 9.** Vertically resolved aerosol composition measured by the ALABAMA, by the HR-ToF-AMS, and by the SP2 (indicated on the top side) during the Arctic air mass period. Note the different scales. Left panel represents the ALABAMA scaled number concentration ($PF \cdot N_{>320}$) with top axis referring to EC-containing (black)- and nss-nitrate (light blue)-containing particles and bottom axis referring to DCA (brown)- and sub-K (green)-containing particles. Right panels represent each median (solid line with marker) and interquartile ranges (shaded area) of the HR-ToF-AMS mass concentrations of sulfate (red), nitrate (blue), ammonium (orange), organic matter (light green), the sulfate-to-organic ratio (pink) as well as the SP2 mass concentration of refractory black carbon (rBC) (black). Uncertainty analyses are given in the Supplement Sect. 7.

range transport (see Sect. 3.2.1). To illustrate this, we show the vertically resolved aerosol composition in Fig. 9 and the aerosol composition as a function of FLEXPART derived PES fraction over different geographical regions in Fig. 13.

During the Arctic air mass period, the ALABAMA analysis suggests that organic matter in the FT was partly composed of particulate DCA. The number concentration of DCA-containing particles was increasing with increasing altitude (Fig. 9). Further, Figs. 10 and 11 show the number concentrations of DCA-containing particles in comparison with the HR-ToF-AMS estimated oxygen-to-carbon (O/C) and hydrogen-to-carbon (H/C) ratios as well as with the FLEXPART derived PES fraction over Arctic regions, respectively. The abundance of DCA-containing particles coincided with air masses with more oxidized organic matter (Fig. 10). Consistent with this result, the size distribution of DCA particles is shifted towards larger diameters compared to other particle types (Fig. 12). Finally, air mass history shows that the abundance of DCA-containing particles correlated with time spent outside Arctic regions (Fig. 11). However, no significant trend with a distinct source region outside the Arctic was found (not shown), as expected from the variety of biogenic and anthropogenic precursor gases as well as primary emissions. These results are in line with findings of an Arctic airborne study by Schmale et al. (2011), who demonstrated the presence of low-volatility highly oxygenated organic aerosol in the summertime Arctic FT that was

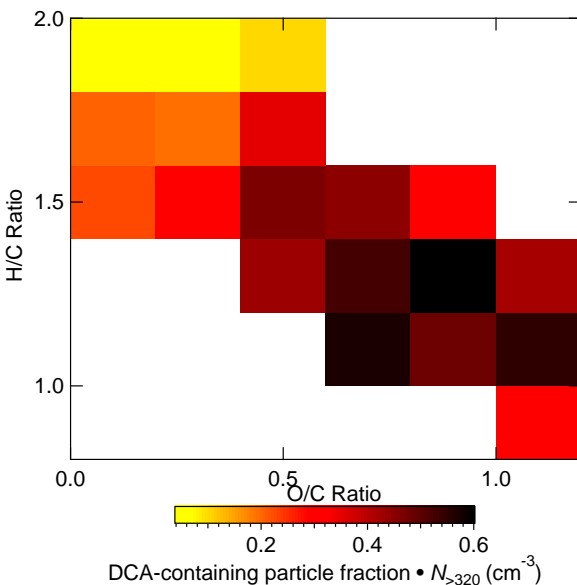

**Figure 10.** Scaled number concentration ($PF \cdot N_{>320}$) of DCA-containing particles measured by the ALABAMA (colored) as a function of the HR-ToF-AMS estimated oxygen-to-carbon (O/C) and hydrogen-to-carbon (H/C) during the Arctic air mass period. Uncertainty analysis is given in Supplement Sect. 7.

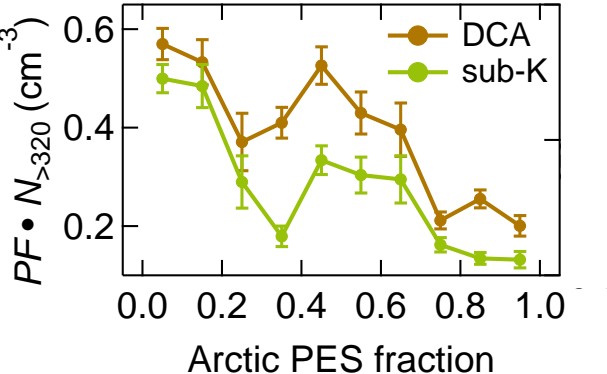

**Figure 11.** Scaled number concentration ($PF \cdot N_{>320}$) of DCA- and sub-K-containing particles measured by the ALABAMA as a function of FLEXPART derived PES fraction over Arctic regions during the Arctic air mass period. Uncertainty analysis is given in the Supplement Sect. 7.

transported over long distances from lower latitudes, irrespective of the source sector and regions. The molecular identity of the observed organic aerosol in Schmale et al. (2011) is not known. Direct observations of DCA in Arctic regions exist,

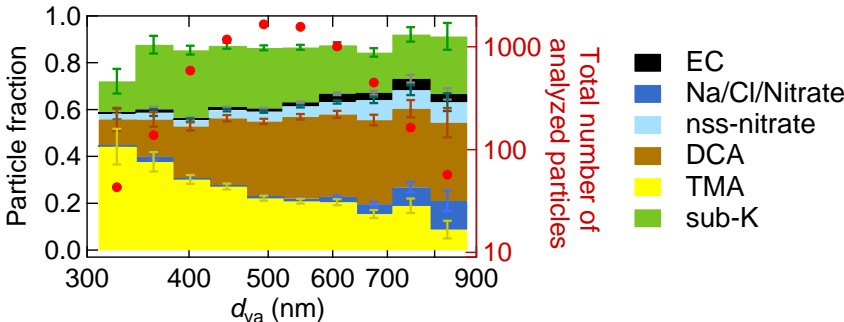

**Figure 12.** Size-resolved aerosol composition of the ALABAMA particle types during the Arctic air mass period. The figure represents the cumulative particle fraction of: EC-containing (black), nss-nitrate-containing (light blue), DCA-containing (brown), Na/Cl/Nitrate-containing (dark blue), TMA-containing (yellow), and sub-K-containing (green) particles as well as the total number of analyzed particles per bin (red). Uncertainty analysis is given in the Supplement Sect. 7

however, up to now confined to measurements in the BL (e.g., Shaw et al., 2010; Kawamura et al., 2012; Leaitch et al.,
2018). These earlier studies suggest summer minimum concentrations of carboxylic acids, due to a combination of diminished transport of precursors and more efficient aerosol wet removal compared to other seasons. We provide new data on the vertical profile of DCA. It is demonstrated that a significant fraction of DCA was present in the summertime Arctic FT, influenced by long range transport from sources outside Arctic regions. Besides the presence in the FT, DCA-containing particles internally mixed with TMA and MSA (see Sect. 3.1) were largely present in the stable stratified BL (Fig. S12). Further, these particles
were detected when the residence time within the Arctic was high (Fig. S13), indicating sources of particulate DCA in the Arctic. This finding is consistent with previous ground-based and shipborne studies (e.g., Kawamura et al., 1996; Kerminen et al., 1999; Kawamura et al., 2010, 2012), partly linking the abundance of DCA in the summertime Arctic to regional sources. Aerosol transport from lower latitudes further influenced Arctic composition by particles that contained nss-nitrate, EC, and potassium. Those particles were more pronounced in the FT compared to the BL (Fig. 9). Air mass history shows contributions
from East Asia and Europe to aerosol composition (Fig. 13). Figure S33 presents an example of the FLEXPART derived transport pathway from East Asia (particularly Japan) to the Arctic. The advection of those air masses to Arctic regions took place mainly at altitudes above 1 km. EC and nss-nitrate measured by the ALABAMA and rBC from the SP2 have their sources in Europe and East Asia (Fig. 13). Air mass history further illustrates that elevated concentrations of EC- and nss-nitrate-containing particles occurred when the residence time over populated and/or industrial areas in Europe and East Asia
was high (compare Fig. 13 with Fig. S14). Consistent with this analysis, the quantitative analysis of rBC by the SP2 shows qualitatively similar trends (Fig. S14). However, rBC mass concentrations were generally low during the NETCARE 2014 study (Schulz et al., 2019), reflecting the largely isolated nature of the summertime Arctic from the rest of the atmosphere. This is in agreement with previous Arctic measurements of BC (or EC) across seasons (e.g., Matsui et al., 2011b; Winiger et al.,

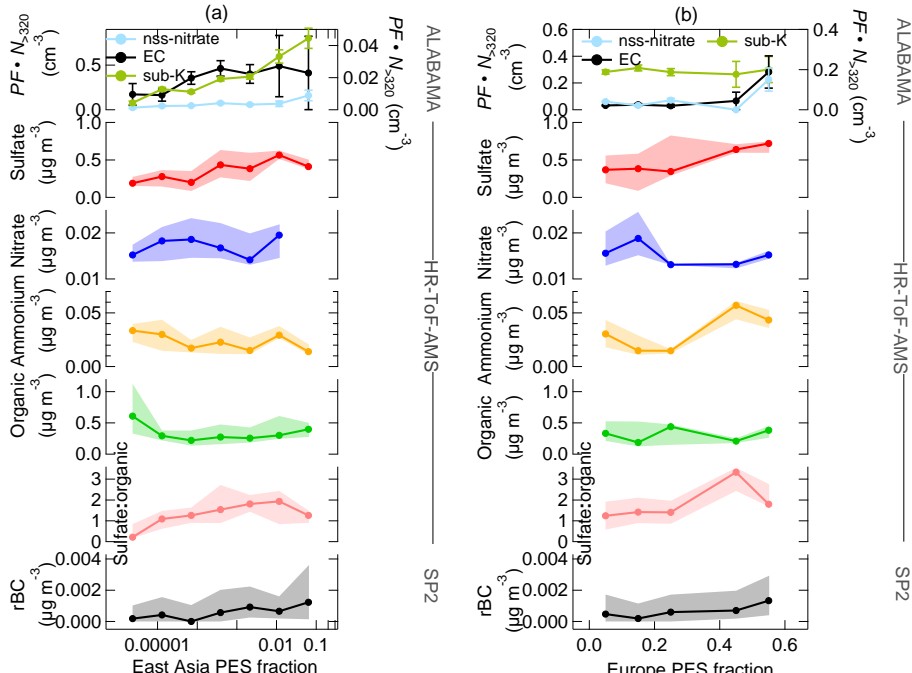

**Figure 13.** Aerosol composition measured by the ALABAMA, by the HR-ToF-AMS, and by the SP2 (indicated on the right side) as a function of FLEXPART derived PES fraction over East Asia (a) and Europe (b) during the Arctic air mass period. Top panel represents each the ALABAMA scaled number concentration ($PF \cdot N_{>320}$) with right axis referring to EC-containing (black) and with left axis referring to nss-nitrate (light blue)- and sub-K (green)-containing particles. Lower panels represent each median (solid line with marker) and interquartile ranges (shaded area) of the HR-ToF-AMS mass concentrations of sulfate (red), nitrate (blue), ammonium (orange), organic matter (light green), and the sulfate-to-organic ratio (pink) as well as the SP2 mass concentration of refractory black carbon (rBC) (black). Uncertainty analyses are given in the Supplement Sect. 7.

2019), demonstrating particularly low concentrations during summer compared to the rest of the year. Matsui et al. (2011b)

could show the influence of Asian anthropogenic emissions on the BC concentration in Arctic summer, in line with our results during the Arctic air mass period. Along with Asian anthropogenic emissions, our results further suggest that anthropogenic sources in East Asia contributed to the abundance of sub-K particles. This particle type correlated with time spent outside Arctic regions (Fig. 11), particularly, with time spent over populated and industrial regions in East Asia (Fig. 13a and Fig. S14). Given that the nss-nitrate, EC, and sub-K particle types were predominantly mixed with secondary substances (sulfate, nitrate, and

oxygen-containing organic matter; Fig. 5c, b, and f, respectively) and that these particle types were larger in diameter compared to BL particles, such as TMA-containing particles (Fig. 12), we conclude a pronounced influence of particle aging processes while particles were transported to Arctic regions.

Measurements by the HR-ToF-AMS combined with air mass history analysis suggest that long range transport from sources outside the Arctic contributed to the abundance of particles containing sulfate and ammonium. Both species show slightly

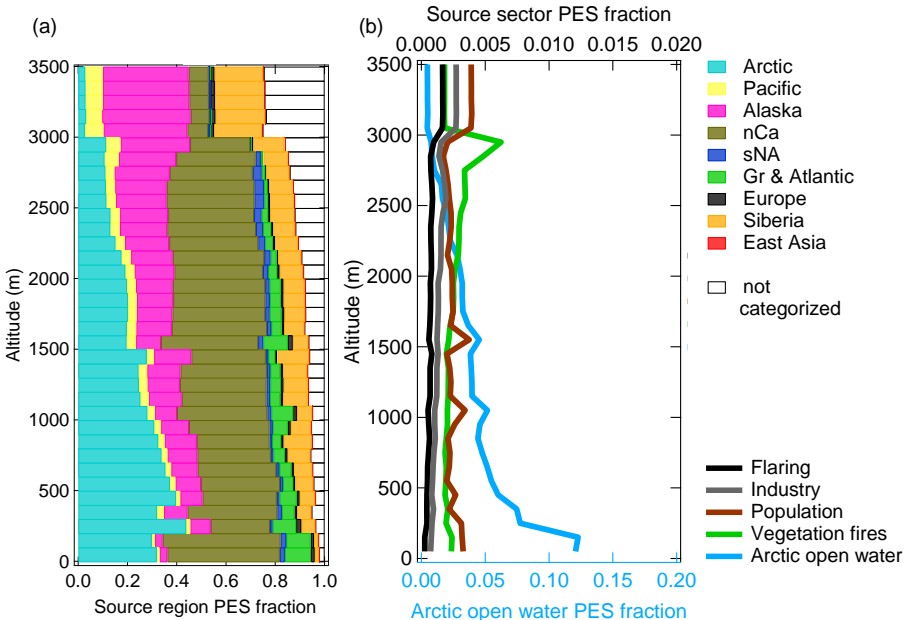

**Figure 14.** Vertically resolved cumulative contribution of (a) source regions and (b) source sectors to sampling altitude during the southern air mass period (July 17-21). For details on flight tracks see Fig. 1. The FLEXPART derived contribution is expressed as a fraction of the potential emission sensitivity (PES) in the model domain lowest vertical level (0‑400 m, except for vegetation fires) over a 15-days backward simulation. A vertical injection layer between 1 and 5 km is applied for vegetation fires (see details in Sect. 2.5.1). Acronyms in (a) are defined as follows: nCa = northern Canada; sNA = southern North America; Gr = Greenland. The white parts depict areas that are not categorized in the selected geographical regions. Bottom and top axes in (b) represent PES fractions of Arctic open water and other source sectors, respectively.

higher mass concentrations in the Arctic FT (Fig. 9). Air mass history suggests that sources in Europe contribute to the presence of particles containing ammonium (Fig. 13b). Sulfate was predominantly abundant when the residence time prior to sampling was high in East Asia and Europe (Fig. 13). In parallel, the sulfate-to-organic ratio was enhanced in the FT (Fig. 9) and coincided with time spent over anthropogenic sources in East Asia and Europe (Fig. 13). It has previously been demonstrated that anthropogenic emissions in northern Eurasia and East Asia contribute to enhanced sulfate concentrations in

the summertime Arctic FT (e.g., Matsui et al., 2011a; Breider et al., 2014; Yang et al., 2018). In line with this, Schmale et al. (2011) observed an increasing sulfate fraction and decreasing organic fraction along with increasing anthropogenic influence from southern latitudes.

### 3.3 Southern air mass period

#### 3.3.1 Meteorological overview and air mass history

Meteorological conditions changed after 12 July due to a low-pressure system that was initially centered to the west above the Beaufort Sea and finally passed through Resolute Bay on 15 July. Until 17 July, flying was impeded by the local weather situation with low visibility, fog, and precipitation along with the low-pressure system (Burkart et al., 2017). However, the measurement region was increasingly impacted by an intense low-pressure system located south of Resolute Bay that led to a different meteorological situation during the period from 17 - 21 July (Bozem et al., 2019). Meteorological conditions were

characterized by overcast sky, occasional precipitation, moist air throughout the lower troposphere, higher wind speeds, and warmer temperatures compared to the Arctic air mass period (Burkart et al., 2017; Bozem et al., 2019). The BL was more difficult to identify compared to the prior period due to a less well defined temperature inversion. According to the detailed BL study by Aliabadi et al. (2016a), the BL height reached altitudes between approximately 200 m and 400 m. However, vertical mixing between the BL and FT was likely promoted caused by a combination of less intense temperature inversions, higher

wind speeds, warmer temperatures, and a higher abundance of clouds compared to the first period (e.g., Stull, 1997; Fuelberg et al., 2010; Tjernström et al., 2012).

This period is referred to as southern air mass period, since air mass history shows the prevalence of air masses originating from southern latitudes. Lagrangian air mass history analysis suggests a pronounced impact of southern latitude sources on Arctic composition (Fig. 14). The cumulative contribution of all regions outside the Arctic dominated air mass history within

480 the lower troposphere (Fig. 14a). By comparing the synoptic situation during the southern air mass period with climatological mean (see Supplement Sect. 3), we found that the presence of the low-pressure system led to a significantly anomalous synoptic situation. This finding explains the discrepancy between our results on air mass history during the second period and results on Arctic transport climatology by Stohl (2006), revealing a largely unperturbed Arctic lower troposphere during summer.

Our results suggest that northern Canada and Alaska exerted the largest influences and the European influence was smaller

than during the first period, consistent with the prevailing cyclonic activity south of Resolute Bay. Transport of air masses from northern Canada was accompanied by emissions from vegetation fires in the Great Slave Lake region (compare Figs. 3a and S36). Contributions from industrial and flaring activities were likely accompanied by transport of air masses from Alaska, Siberia, southern North America, and Europe (Fig. 14). Also, the contribution of air masses in the lower FT that had resided above Arctic open water was significantly lower during the southern air mass period compared to the Arctic air mass period

(compare Figs. 6b and 14b). Regions that were not categorized (Mexico and northern Africa both north of 25° N, see Fig. 2) contribute significantly to air mass history at altitudes above 3 km (Fig. 14a). However, our data on aerosol composition at altitudes above 3 km are sparse, with the result that our further data analysis is limited to altitudes below 3 km.

#### 3.3.2 Vegetation fire and anthropogenic influences on particle composition

Our measurements combined with air mass history analysis suggest that long range transport of vegetation fire emissions con-

495 tributed to the abundance of particles containing nss-nitrate, ammonium, organics, and sulfate in the summertime Arctic lower

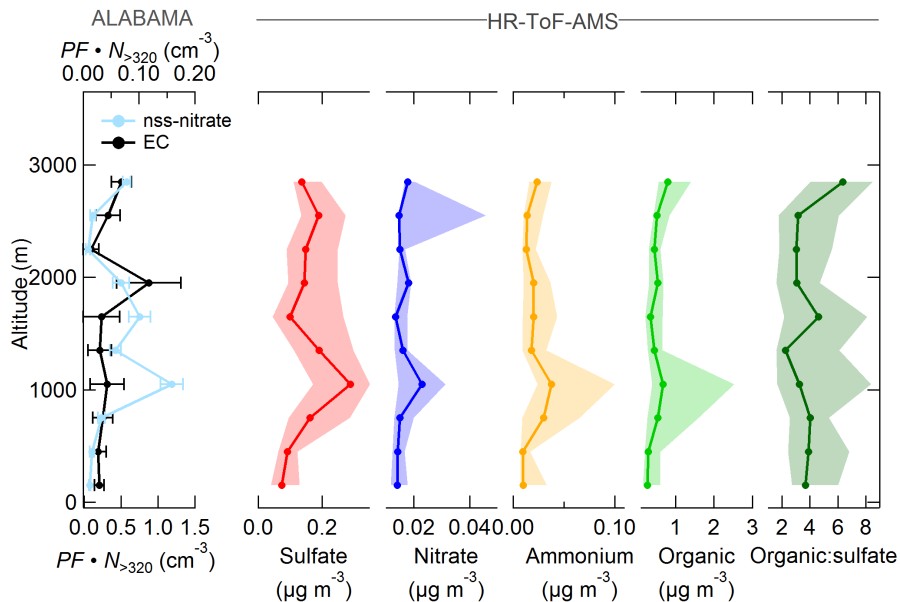

**Figure 15.** Vertically resolved aerosol composition measured by the ALABAMA and by the HR-ToF-AMS (indicated on the top side) during the southern air mass period. Note the different scales. Left panel represents the ALABAMA scaled number concentration ($PF \cdot N_{>320}$) with top axis referring to EC-containing (black) and bottom axis referring to nss-nitrate (light blue)-containing particles. Right panels represent each median (solid line with marker) and interquartile ranges (shaded area) of the HR-ToF-AMS mass concentrations of sulfate (red), nitrate (blue), ammonium (orange), organic matter (light green), and the organic-to-sulfate ratio (dark green). Uncertainty analyses are given in the Supplement Sect. 7.

FT. Figure 15 illustrates the vertically resolved aerosol composition during the southern air mass period. We can show that those species were more pronounced in the FT compared to the BL, suggesting the influence of long range transport on aerosol composition. Comparing both periods, the nitrate, ammonium, and organic concentrations measured by the ALABAMA and HR-ToF-AMS were higher during the southern air mass period (compare Figs. 9 and 15), reflecting the pronounced influence

of vegetation fires during this time. We used the CO mixing ratio to demonstrate the contribution of combustion sources on aerosol composition (Fig. 16). Additionally, Figs. 17 and 18 provide the aerosol composition as a function of FLEXPART derived PES fraction over different source sectors (including vegetation fires) and over different source regions, respectively. The number concentration of nss-nitrate particles measured by the ALABAMA increased with elevated CO mixing ratios (Fig. 16). In parallel, air mass history shows that nss-nitrate particles were largely abundant when time spent over vegetation fires

was high (Fig. 17a). Given that the majority of nss-nitrate-containing particles were internally mixed with potassium (see Sect. 3.1), we have additional indications for their biomass burning origin (Silva et al., 1999; Hudson et al., 2004; Pratt and Prather, 2009; Pratt et al., 2011; Quinn et al., 2002; Schill et al., 2020). The inclusion of nitrate and sulfate with potassium is indicative for atmospheric processing of biomass burning particles while transported into the Arctic.

Consistent with these results, nitrate mass concentrations measured by the HR-ToF-AMS increased with increasing influence

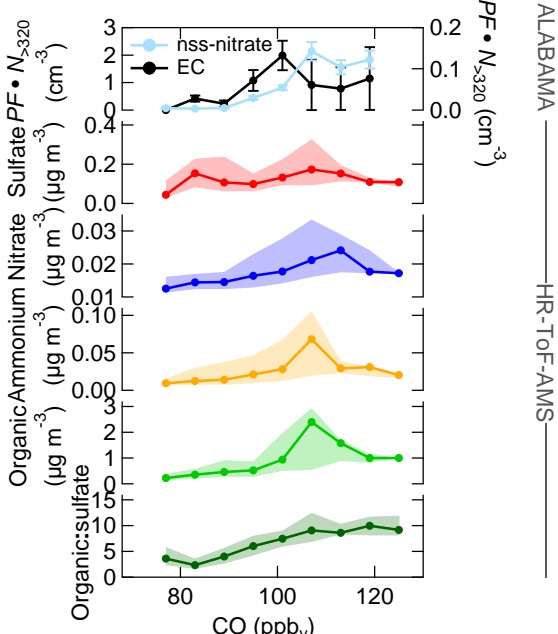

**Figure 16.** Aerosol composition measured by the ALABAMA and by the HR-ToF-AMS (indicated on the right side) as a function of CO mixing ratios during the southern air mass period. Top panel represents the ALABAMA scaled number concentration ($PF \cdot N_{>320}$) with right axis referring to EC-containing (black) and left axis referring to nss-nitrate-containing (light blue) particles. Lower panels represent each median (solid line with marker) and interquartile ranges (shaded area) of the HR-ToF-AMS mass concentrations of sulfate (red), nitrate (blue), ammonium (orange), organic matter (light green), and the organic-to-sulfate ratio (dark green). Uncertainty analyses are given in the Supplement Sect. 7.

from vegetation fires (Fig. 17a). Similar to nitrate, enhanced sulfate and ammonium concentrations measured by the HR-ToF-AMS coincided with both high CO mixing ratios and with time spent over vegetation fires (Figs. 16 and 17a). Along with vegetation fires and high CO mixing ratios, the organic-to-sulfate ratio was increasing, indicating the pronounced influence of vegetation fire emissions on the presence of organic matter in the summertime Arctic FT. Finally, particles containing nss-nitrate, sulfate, ammonium, and organics were present when the residence time over northern Canada was high (Fig. 18a),

suggesting that vegetation fires in northern Canada together with the low-pressure system south of Resolute Bay contributed to aerosol composition in the summertime Arctic FT. However, Figure 16 further demonstrates that sulfate, nitrate, ammonium, and organic matter measured by the HR-ToF-AMS show decreasing concentrations with highest CO mixing ratios. Whereas the number fraction of nss-nitrate-containing particles and the organic-to-sulfate ratio (Fig. S22 and Fig. 16, respectively) were increasing with high CO mixing ratios. Along with the low-pressure system, the transport of air masses that were characterized

by high CO mixing ratios was accompanied with precipitation events, as can be seen in GFS archive data (Wetter3, 2020). It is therefore conceivable that this transport pathway was linked to intensive particle wash-out events (Garrett et al., 2010, 2011; Matsui et al., 2011a; Browse et al., 2012; Sato et al., 2016).

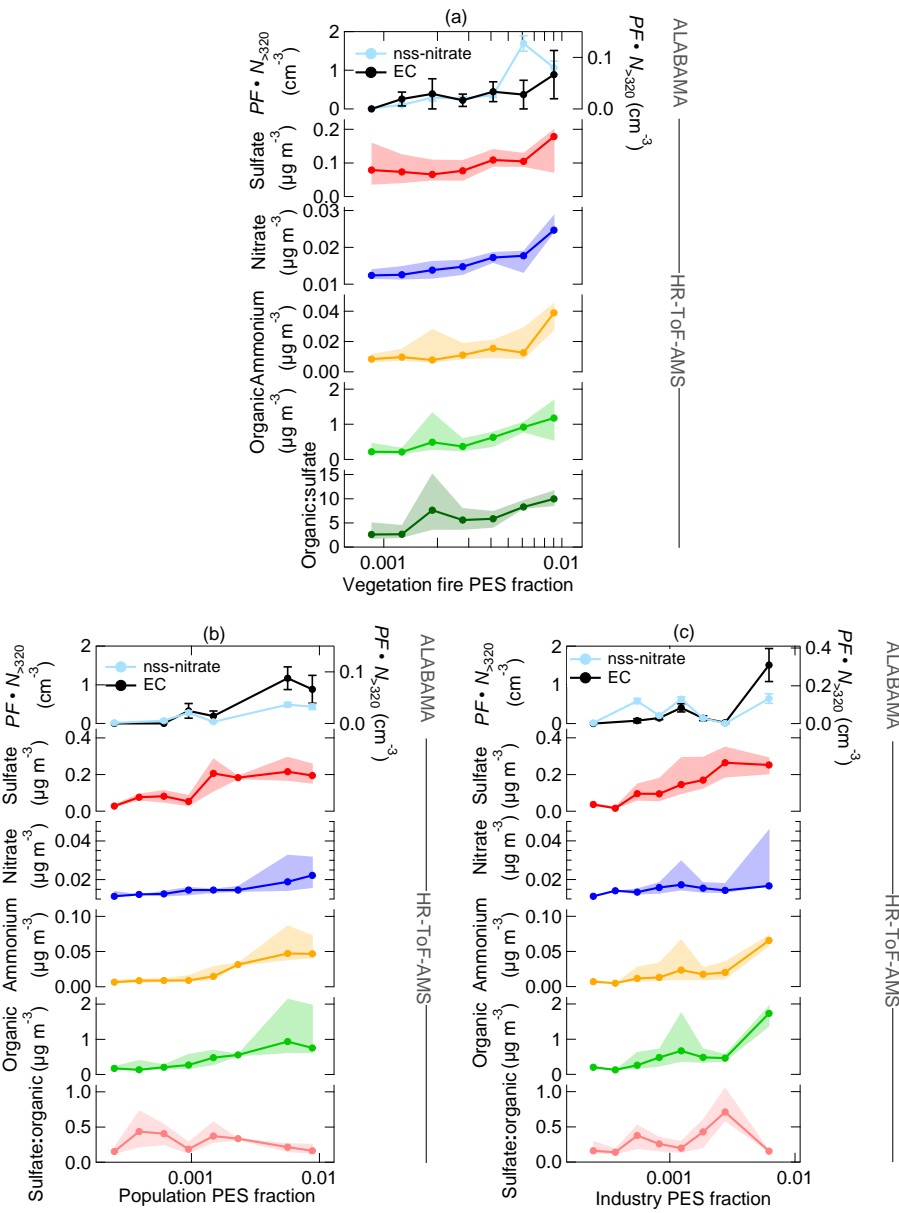

**Figure 17.** Aerosol composition measured by the ALABAMA and by the HR-ToF-AMS (indicated on the right side) as a function of FLEXPART derived PES fraction over (a) vegetation fires, (b) populated areas, and (c) industrial areas during the southern air mass period. Top panel represents the ALABAMA scaled number concentration ($PF \cdot N_{>320}$) with right axis referring to EC-containing (black) and left axis referring to nss-nitrate-containing (light blue) particles. Lower panels represent each median (solid line with marker) and interquartile ranges (shaded area) of the HR-ToF-AMS mass concentrations of sulfate (red), nitrate (blue), ammonium (orange), and organic matter (light green) as well as the organic-to-sulfate ratio (dark green) and the sulfate-to-organic ratio (pink). Uncertainty analyses are given in the Supplement Sect. 7.

Earlier measurements show that background concentrations of ammonium and nitrate are generally low in Arctic summer (Quinn et al., 2002; Kuhn et al., 2010; Chang et al., 2011; Schmale et al., 2011; Quennehen et al., 2011; Hamacher-Barth et al.,

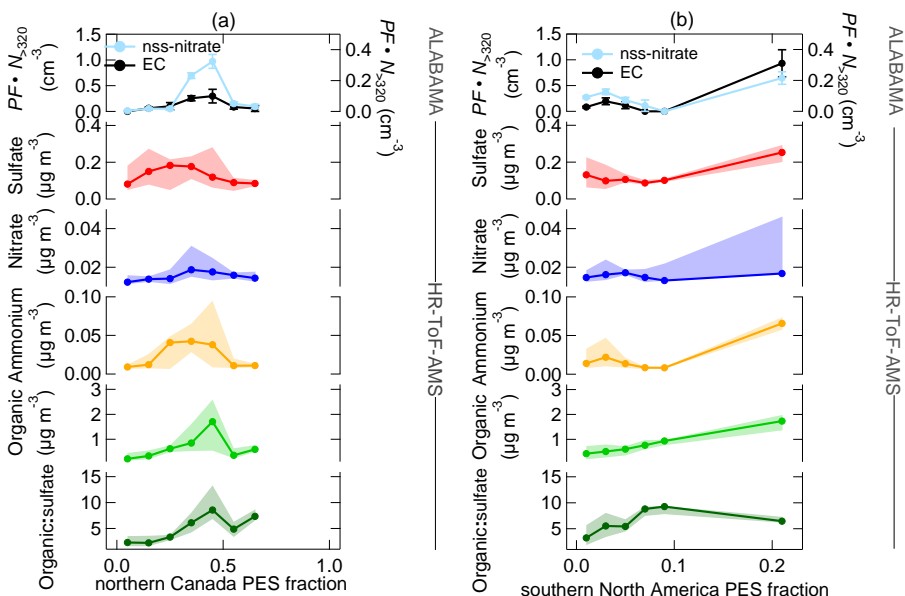

**Figure 18.** Aerosol composition measured by the ALABAMA and by the HR-ToF-AMS (indicated on the right side) as a function of FLEXPART derived PES fraction over northern Canada (a) and southern North America (b) during the southern air mass period. Top panel represents each the ALABAMA scaled number concentration ($PF \cdot N_{>320}$) with right axis referring to EC-containing (black) and left axis referring to nss-nitrate-containing (light blue) particles. Lower panels represent each median (solid line with marker) and interquartile ranges (shaded area) of the HR-ToF-AMS mass concentrations of sulfate (red), nitrate (blue), ammonium (orange), organic matter (light green), and the organic-to-sulfate ratio (dark green). Uncertainty analyses are given in the Supplement Sect. 7.

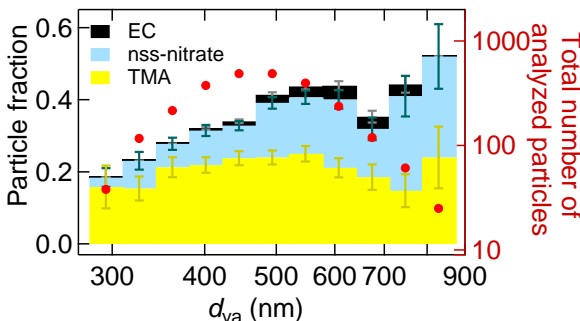

**Figure 19.** Size-resolved aerosol composition of the ALABAMA particle types during the southern air mass period. The figure represents the cumulative particle fraction of: EC-containing (black), nss-nitrate-containing (light blue), and TMA-containing (yellow) particles as well as the total number of analyzed particles per bin (red). Uncertainty analysis is given in the Supplement Sect. 7

2016; Lange et al., 2018), which we can confirm under the pristine conditions during the Arctic air mass period (Fig. 9). How-

ever, our vertical profile measurements during the southern air mass period (Fig. 15) show that episodic and localized transport of vegetation fire emissions can perturb background concentrations of nitrate, ammonium, and organic matter. This finding is in line with modeling and observational studies (Hecobian et al., 2011; Brock et al., 2011; Lathem et al., 2013; Kondo et al., 2011; Matsui et al., 2011a; Breider et al., 2014), reporting on the response of nitrate, ammonium, and organic concentrations in the Arctic on the transport of large fire emissions from sub-Arctic regions. Also gas-phase measurements demonstrate the large input of ammonia (precursor gas of particulate ammonium) by vegetation fire emissions to the summertime Arctic lower troposphere (Lutsch et al., 2016).

The mass concentration of rBC measured by the SP2 (during three out of five flights) provides evidence that those vegetation fires contributed also to the presence of rBC in the summertime Arctic FT (see Fig. S15). This result is in line with earlier studies, analyzing the source attribution of BC (or EC) in the summertime Arctic and suggesting a dominant influence of biomass burning on the BC (or EC) burden in summer (e.g., Hirdman et al., 2010; Bourgeois and Bey, 2011; Stohl et al., 2013; Breider et al., 2014; Xu et al., 2017; Sobhani et al., 2018; Winiger et al., 2019; Zhu et al., 2020). In contrast, the ALABAMA analysis shows no significant evidence of influences from vegetation fires on the abundance of EC particles, which might be explained by the following reasons. It is known from previous field studies that chemically aged BC particles from vegetation fires are thickly coated with organics and inorganics (e.g., Paris et al., 2009; Singh et al., 2010; Kondo et al., 2011; Brock et al., 2011). In addition, a laboratory study by Silva et al. (1999) reported the majority of biomass burning particles to include potassium. Given that potassium is known to produce matrix effects, owing to its low ionization potential (e.g., Silva and Prather, 2000), positive ion signals, other than $K^+$, in biomass burning particle spectra are likely suppressed by the presence of potassium. Further, Moffet and Prather (2009) reported negative ion mass spectra of aged soot to be dominated by nitrate and sulfate signals. Together, it is likely that signals of carbon cluster ions, indicative for the presence of EC/soot, are suppressed by other substances that are abundant in biomass burning particles. This hypothesis is consistent with previous SPMS measurements during field experiments at southern latitudes, characterizing biomass burning aerosol (e.g., Pratt et al., 2011; Zauscher et al., 2013; Gunsch et al., 2018). These earlier studies show that mean mass spectra of aged biomass burning particles are generally characterized by the concurrent presence of potassium, sulfate, nitrate, organic acids, and nitrogen-containing organics, confirming findings in our study (see nss-nitrate in Figs. 5c and 17a).

Besides the influence of vegetation fires, the presence of sulfate, ammonium, nitrate, EC, and organic matter in the summertime Arctic FT can further be linked to anthropogenic emissions from southern latitudes. Air mass history illustrates the coincidence of enhanced sulfate, ammonium, nitrate, and organic concentrations with time spent over populated and/or industrial areas (Fig. 17b-c). In addition, those species were predominantly abundant when the residence time prior to sampling was high in southern North America (Fig. 18b). Along with this, highest sulfate concentrations were observed at maximum PES fraction over anthropogenic sources compared to over vegetation fires (Fig. 17). Consistently, it has previously been shown that the sulfate burden in the summertime Arctic FT is dominated by transport of a wide variety of anthropogenic emissions in northern Eurasia, North America, and East Asia (e.g., Shindell et al., 2008; Hirdman et al., 2010; Kuhn et al., 2010; Bourgeois and Bey, 2011; Schmale et al., 2011; Matsui et al., 2011a; Breider et al., 2014; Yang et al., 2018; Sobhani et al., 2018). Interestingly, we could not observe a significant trend for the sulfate-to-organic ratio as a function of anthropogenic sources (Fig. 17b-c),

indicating the importance of anthropogenic emissions on the organic fraction in the Arctic FT.

The ALABAMA data further show that the presence of EC particles in the Arctic FT can partly be associated with industrial emissions in southern North America (compare Figs. 17c and 18b), likely including flaring emissions (compare Fig. S34 with Fig. 3b-c). Figure S34 illustrates that long range transport from southern North America to Arctic regions might be accompanied by emissions from oil production industry in Alberta, Canada (compare with Fig. 3c). This result is consistent with recent modeling studies, demonstrating the importance of anthropogenic BC sources in northern Eurasia and North America (e.g., Huang et al., 2010; Bourgeois and Bey, 2011; Sobhani et al., 2018). Particularly, studies by Stohl et al. (2013) and Evangeliou et al. (2018) demonstrated the importance of distant sources in southern North America to affect the abundance of EC-containing particles in the Arctic summer. The authors showed that regions in western Canada (Alberta), operating numerous oil producing facilities, influenced aerosol particle composition in the Arctic summer. However, contributions from sources in South Asia cannot be confirmed by our measurements, which might be related to maximum sampling altitude up to 3.5 km that is less sensitive to Asian pollution sources (e.g., Bourgeois and Bey, 2011; Sharma et al., 2013; Sobhani et al., 2018). Finally, nss-nitrate- and EC-containing particles were larger in diameter compared to BL particles, such as TMA-containing particles (Fig. 19), consistent with pronounced influences of aerosol processing during long range transport. Taken together, summertime Arctic FT aerosol particle composition under the influence of a prevailing low-pressure system was driven by the transport of emissions from vegetation fires and anthropogenic sources in northern Canada and southern North America.

## 4 Conclusions

Motivated by the limited knowledge of summertime Arctic aerosol, our study aimed to improve our understanding of sources impacting the vertical structure of aerosol particle chemical composition and mixing state. Aircraft-based measurements were performed using two complementary aerosol mass spectrometers, the ALABAMA and HR-ToF-AMS, and an SP2 in the summertime Canadian Arctic lower troposphere.

Our measurements and air mass trajectory analysis show that the vertical structure of aerosol particle composition in the summertime Arctic lower troposphere was driven by a combination of aerosol processes occurring along northward transport pathways and within Arctic regions. In our prior studies, we could show that particle composition in the isolated and stable stratified boundary layer was characterized by the abundance of MSA, sodium, chloride, and organic matter that is partly associated with particulate TMA (Willis et al., 2016, 2017; Köllner et al., 2017). Further, we observed that TMA and sea spray particles were externally mixed and smaller in diameter, suggesting a role of marine-derived secondary organic aerosol formation occurring alongside with primary sea-to-air emissions. Here, the presented results confirm the marine source of TMA by our FLEXPART air mass history analysis. The presence of TMA-containing particles correlated, similar to sea spray particles, with time spent over Arctic open water regions. Overall, this study together with findings from earlier work on the NETCARE 2014 measurements emphasize the importance of marine SOA formation on the vertical structure of particle composition in the summertime Arctic (Willis et al., 2016, 2017; Köllner et al., 2017; Croft et al., 2019).

As new results compared to our prior studies, we discussed particle composition and mixing state measured during two distinc-

tive synoptic situations in the summertime Arctic lower FT, namely the Arctic and the southern air mass periods. In contrast to the isolated Arctic BL, particle composition aloft was characterized by the presence of nss-nitrate, EC, ammonium, sulfate, and oxidized organics that can partly be attributed to DCA transported over long distances from mid-latitude sources. However, particle composition in the FT varied with the synoptic situations, owing to different particle sources and transport pathways. Comparing both periods, the organic, ammonium, and nitrate concentrations measured by HR-ToF-AMS as well as the nitrate fraction by the ALABAMA were higher during the southern air mass period, reflecting the pronounced influence of vegetation fires in northern Canada during this time and of transport associated with the prevailing low-pressure system. Our air mass history analysis showed that elevated concentrations of nitrate, ammonium, rBC, and organic matter occurred when the residence time over vegetation fires in northern Canada was high. Further, those particles were largely present in air masses with high CO mixing ratios. Also the organic-to-sulfate ratio coincided with time the air mass spent over these fires.

Besides the influence of vegetation fires, anthropogenic sources in Europe, North America, and East Asia contributed to the presence of particles containing sulfate, nitrate, EC, organic matter, and ammonium in the summertime Arctic FT. The presence of those particles correlated with time spent over populated and industrial areas in Europe, East Asia, and southern North America. Measurements of enhanced EC concentrations can be traced back to oil production industry, including flaring, in Alberta (Canada), by means of the FLEXPART analysis. We further have evidence that EC and nss-nitrate particles present in the Arctic FT were exposed to chemical aging processes during northward transport. First, the size distributions of these particle types are shifted towards larger diameters compared to BL particles, such as TMA-containing particles. Second, these particles types were largely internally mixed with secondary substances, such as sulfate, nitrate, and oxygen-containing organic matter.

Particularly during the first period, our analysis showed that the majority of DCA-containing particles were observed in the Arctic FT and in air masses that spent significant time outside Arctic regions. Long range transport of aerosol and precursors from distant sources thus influenced the abundance of DCA particles in the summertime Arctic FT. These findings add to our understanding of organic acids in summer Arctic aerosol, because previous studies were confined to ground-based and ship-borne datasets. Combined with the stable stratified nature of the Arctic lower troposphere, previous BL datasets were therefore less sensitive to organic acids from long range transport and related processes.

Together, these results indicate the important role of mid-latitude aerosol sources on summertime Arctic particle composition in the lower FT. Particularly, contributions from biomass burning emissions on summertime Arctic aerosol are likely to increase under scenarios for the future climate (e.g., Randerson et al., 2006; de Groot et al., 2013; Marelle et al., 2018; Evangeliou et al., 2019). To our knowledge, this is the first study providing vertically resolved particle chemical composition analysis in the summertime Arctic FT, by using two complementary aerosol mass spectrometers with black carbon measurements from an SP2. However, open questions remain, partly owing to limitations of the current study. Being limited by the spatial and temporal range of the NETCARE 2014 measurements, it is difficult to assess the representativeness of the findings for the whole Arctic. Future widespread and long-term measurements would help to extend the results to other Arctic regions and seasons. Lagrangian flight experiments would help to better constrain the relative role of aerosol processing, including wet removal and photochemical aging, on summertime Arctic particle composition. Based on limitations in sampling altitude, future Arctic air-

borne measurements in the mid to upper troposphere are necessary to further characterize the impact of mid-latitude pollution
sources on summertime Arctic aerosol properties. Finally, it is of relevance to further characterize organic compounds in Arctic aerosol along with their sources, formation processes, and impact on clouds.

*Data availability.*   The ALABAMA and FLEXPART data can be accessed by contacting the corresponding author F. Köllner (f.koellner@mpic.de) or the second author J. Schneider (johannes.schneider@mpic.de). Other NETCARE data presented in this publication are publicly available through the Government of Canada open data portal (https://open.canada.ca).

*Author contributions.*   JA, RL, and AH designed the research project. FK, JS, HB, RL, MW, and JB carried out the measurements. AA processed the AIMMS-20 data. TK, FH, and JS re-designed and further developed the ALABAMA for aircraft-based measurements. DK provided the FLEXPART output. MW provided the HR-ToF-AMS data. HS provided the SP2 and UHSAS data. HB provided the CO data. FK performed the data analyses with critical feedback from JS, PH, MW, TK, DK, and JA. FK wrote the manuscript. The manuscript was critically reviewed by JS, PH, SB, JA, MW, RL, DK, and HS.

*Competing interests.*   The authors declare that they have no conflict of interest.

*Acknowledgements.*   The authors thank Kenn Borek Air Ltd., in particular our pilots Kevin Elke and John Bayes, as well as our aircraft maintenance engineer Kevin Riehl. We thank Jim Hodgson and Lake Central Air Services in Muskoka, Jim Watson (Scale Modelbuilders, Inc.), Julia Binder and Martin Gehrmann (Alfred Wegener Institute, AWI) for their support of the integration of the instrumentation in the aircraft. We thank Bob Christensen (University of Toronto), Lukas Kandora, Manuel Sellmann, Christian Konrad, and Jens Herrmann
(AWI), Desiree Toom, Sangeeta Sharma (ECCC), Kathy Law and Jenny Thomas (LATMOS) for their support before and during the study. We thank Christiane Schulz (MPIC) for her support during the integration of the instruments in Muskoka. We thank the Biogeochemistry department of MPIC for providing the CO instrument and Dieter Scharffe for his support during the preparation phase of the campaign. We thank the Nunavut Research Institute and the Nunavut Impact Review Board for licensing the study. Logistical support in Resolute Bay was provided by the Polar Continental Shelf Project (PCSP) of Natural Resources Canada under PCSP Field Project 218-14. Funding
for this work was provided by the Natural Sciences and Engineering Research Council of Canada through the NETCARE project of the Climate Change and Atmospheric Research Program, the Alfred Wegener Institute, Environment and Climate Change Canada, and the Max Planck Society. Parts of this research were conducted using the supercomputer Mogon and/or advisory services offered by Johannes Gutenberg University Mainz (hpc.uni-mainz.de), which is a member of the AHRP (Alliance for High Performance Computing in Rhineland Palatinate, www.ahrp.info) and the Gauss Alliance e.V. The authors gratefully acknowledge the computing time granted on the supercomputer
Mogon at Johannes Gutenberg University Mainz (hpc.uni-mainz.de). ECMWF data were retrieved from the MARS archive and ECMWF is acknowledged for the data provision. Special thanks to the whole NETCARE-team for data exchange, discussions and support. The authors thank Alexandre Caseiro (MPIC) for his help and discussions on satellite data and for providing the VIIRS nightfire "*Flares Only*" product.

We thank Oliver Appel (MPIC) for helpful discussions on data analysis of the ALABAMA instrument. We thank two anonymous referees for helpful suggestions that improve the manuscript.

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
