# Peer review of "Chemical composition and source attribution of submicron aerosol particles in the summertime Arctic lower troposphere"

_Atmospheric Chemistry and Physics, 2020_

## Referee Comment (RC1) · Anonymous Referee #1 · 31 Aug 2020

Köllner et al. provide a detailed summary on Arctic boundary layer and lower free tropospheric aerosol composition during the summer using multiple analytical methods, satellite data, and air mass trajectory analyses. Two main air sectors were defined during their observations including Arctic air from the north and southerly transport from lower latitudes. Sources reported include wildfire, industrial, and other combustion sources in addition to marine sources such as sea spray. While this detailed information fills a key gap in aerosol observations during the Arctic summertime, there are a few issues that should be addressed prior to publication as delineated below.

The abstract is essentially a list of findings and contains no information on how the current work ties into broader implications for Arctic climate or why these results are novel. The authors should consider adding a couple sentences to demonstrate the importance of their work.

The authors provide a very comprehensive, detailed account of their results, but how their findings fit into the bigger picture is not apparent. A synopsis of *why* assessing the aerosol composition to this level of detail is needed to provide a clear picture on the importance of such measurements. For example, why do we care about knowing the abundance of DCA, sulfate, BC, and organic aerosols, specifically? What has previous work elucidated in terms of these specific aerosols and their impacts on radiation and cloud formation? What observations exist and why are the limited? What are models lacking that require data such as those from the current work? These are key questions for Arctic aerosol-cloud-radiation studies that should be elaborated upon to motivate the purpose of this work. Surely, any Arctic aerosol scientist would know why this work is important; however, to emphasize the importance to a larger crowd (and, say, a cloud physicist looking for information on why aerosols are important for clouds), a broader context and implications should be included in the introduction and revisited in the conclusions.

While the authors do compare their findings a couple of times to Schmale et al. (2011) and reference others like Shaw et al. (2010), Kawamura et al. (2012), and Leaitch et al. (2018), the comparisons are quite limited, even to the extent where they state "This finding is consistent with previous ground-based and shipborne studies..." and only provide references but not what the findings from those studies. In order to put their results into the context of others, and demonstrate why their results fill key observational gaps, a mentioning of *how* their results compare to other analogous studies by briefly describing what those studies concluded is needed. Also, other key studies such as those by Quinn et al. (2002, 2009) and Winiger et al. (2019), to name a few, that describe aerosol composition at various sites throughout the Arctic should be included in the discussion and comparison of results. Specifically, for the air mass source assessment, a more direct link to Stohl (2006) is warranted, given that study evaluated air mass sources of the Arctic using FLEXPART in detail. Are the current results consistent or contradictory to previous work?

Quinn, P. K., Miller, T. L., Bates, T. S., Ogren, J. A., Andrews, E., and Shaw, G. E.: A 3-year record of simultaneously measured aerosol chemical and optical properties at Barrow, Alaska, Journal of Geophysical Research: Atmospheres, 107, AAC 8-1-AAC 8-15, 10.1029/2001JD001248, 2002.

Quinn, P. K., Bates, T. S., Schulz, K., and Shaw, G. E.: Decadal trends in aerosol chemical composition at Barrow, Alaska: 1976–2008, Atmos. Chem. Phys., 9, 8883–8888, https://doi.org/10.5194/acp-9-8883-2009, 2009.

Winiger, P., Barrett, T. E., Sheesley, R. J., Huang, L., Sharma, S., Barrie, L. A., Yttri, K. E., Evangeliou, N., Eckhardt, S., Stohl, A., Klimont, Z., Heyes, C., Semiletov, I. P., Dudarev, O. V., Charkin, A., Shakhova, N., Holmstrand, H., Andersson, A., and Gustafsson, Ö.: Source apportionment of

circum-Arctic atmospheric black carbon from isotopes and modeling, Science Advances, 5, eaau8052, 10.1126/sciadv.aau8052, 2019.

The results and discussion section seems a bit fragmented as the discussion goes back and forth between the north and south influences, and describing single particle composition for both is combined. Perhaps the entire section would be easier to follow if all data (composition, FLEXPART, and satellite data) were discussed in tandem but organized by source region (north versus south). Also, why are vegetation fires and anthropogenic sources a separate section? Aren't they technically a south air mass influence sources and aren't they technically long-range transported? Not clear why these are segregated from the section on the southerly sources.

Seems like Figure S1 should be in the paper. There is a description of the site locations in text, but it is not clear to those unfamiliar with the area. I would think this is important to show for a clear link between sources and transport pathways, and the type of land they cover.

It was not clear to me until later in the methods why the UHSAS size range was restricted. Please provide the size range of the ALABAMA in section 2.2.1. Also, what is the size range of the SP2 data used?

The statement on the top of page 16 indicates the aerosol data presented are from "sampling outside of clouds". But does that mean the measurements were conducted during cloud free periods or were the aerosols sampled below/above clouds during cloudy periods, or some combination of each? I assume based on the generalized north and south air mass descriptions that clouds were mostly present during the latter half of the study, but it is not clear when exactly the aerosols were sampled during cloudy versus clear conditions. This would be important to discuss, at least briefly, because aerosol composition can be quite different in the Arctic boundary layer during clear versus cloudy conditions.

What is the relative contribution of the Arctic versus southern air mass periods? It is obvious for the beginning and end, but what about the 5-day transitional period? The authors could provide the % of each air mass contribution for the entire study at the bottom of page 11.

Why is organic:sulfate only shown in select panels/figures (i.e. Figures 10, 11, 14, and 15)? The ratio is also shown as a different color in Figure 11a but should be consistent with the rest of the figures.

For Figure 15, there does not seem to be major differences between the PES sources for fires, population, and industry. The authors describe each separately, but some discussion on why they look so similar is needed.

---

## Referee Comment (RC2) · Anonymous Referee #2 · 22 Sep 2020

Kollner et al present summertime aircraft-based online aerosol chemical composition data from two mass spectrometers, providing both single-particle and bulk composition information, which is quite unique. These novel data were collected as part of the Canadian NETCARE campaign and significantly contribute to understanding of Arctic summertime aerosol composition, knowledge of which is observationally limited, especially when considering non-ground-based measurements. This is an important dataset to publish. Overall, the paper is well-done and contributes significantly to the field. The vertical profiles of aerosol composition and source region analyses using backward air mass trajectories are a good analysis approach to correlate composition with air mass source regions. It is excellent that the authors scaled the ALABAMA

data to number concentrations and conducted uncertainty analysis. The supplementary material is comprehensive. However, I have one major data analysis concern, as described below, about the significant fraction of unexamined single-particle data, the omission of which potentially impacts the presentation and interpretation of the results. In addition, as described below, the authors' definition of "Arctic" as north of 73.5 deg is inconsistent with its geographic definition and norms of other studies, and impacts the source region attribution and discussion. Otherwise, the majority of my other comments are minor.

My major data analysis concern regards the approach in the classification of the AL-ABAMA single-particle mass spectra. As stated on lines 140-141, "the ion marker method classified particles based on the presence of pre-selected species that are of interest." This means that the authors have chosen what chemical species to focus on, which is not necessarily problematic; however, only 54% of the single-particle mass spectra were classified. This means, as shown in Figure 6 and stated in the Figure S9 caption and on line 320, that 46% of the single-particle mass spectra are not included in the discussed data/results. Being nearly half of the ALABAMA data and the fact that this paper is meant to be an overview/summary of the results during NETCARE, I am concerned that the authors may inadvertently be biasing their results by not classifying the remaining data, and I do not consider 46% to be an acceptable fraction of unclassified/real spectra. Figure S9 shows that at least the majority of these mass spectra are real, and therefore, they need to be classified and discussed. Figure 6 shows that this requires further ion marker searches to classify the particles, and a good starting point for this is the examination of the mass spectra shown in Figure S9. In their previous paper (Kollner et al 2017, ACP), which focused on trimethylamine-containing particles, they included an additional particle type called "K-containing" (potassium-containing). Figure S9 shows the mean mass spectrum of the "Others" (non-classified) particles, and this shows a dominant K+ peak, carbonaceous ions, and sulfate. Figure S9 is also nearly identical to the mean K-containing mass spectra in Figure 4e of Kollner et al (2017, ACP). Since this particle type is consistent with biomass burning (the mass

spectrum of which is discussed on lines 441-444), it seems even more important to include a K-containing, or similar, particle type to classify more of the remaining data. Given the similarity of Figure S9 to biomass burning, it is necessary for these data to be included in the evaluation of the vegetation fire influence portion of the manuscript in particular. How does the PF.N>320 vs vegetation fire PES fraction compare for when all K-containing particles are included, or how does it compare currently for this "Others" category as a starting point? Conducting a K+ search on the remaining mass spectra will also be informative to then examine the mean mass spectra of the still unclassified particles, which can be further classified to potentially unearth another particle type that may provide greater insights. It is possible that by including more of the available data that some of the presented trends may change, or new trends may emerge, and without including more of the available data, I am concerned about the results being biased by the chosen ion markers.

Section 2.5.2 & Figure 3: Why does the category of "Arctic open water" not include the full Arctic Ocean, including the Beaufort and Chukchi Seas? Figure S36 shows some surface influence from this region, so inclusion of only open water north of 73.5 deg means that additional Arctic open water is instead classified as "Alaska" and missed in the "Arctic open water" source sector analysis. The authors could sub-categorize the High Arctic, but the full Arctic Ocean should be considered as "Arctic" for consistency with its geographic definition and for comparison to other work. This is also true with respect to the definition of 73.5 deg north as "Arctic", when the Arctic Circle is at 66 deg 34' N. Given little population north of the Arctic Circle, it would seem more appropriate to either sub-divide the Arctic into two categories, or include 66-73 N in the Arctic category. The categories of Alaska, northern Canada, Europe, Greenland & the Atlantic, and Siberia extend much, much further north, and therefore, air mass influence still in the "Arctic" (i.e. >66 deg N) would be discussed as being from a further south influence with the categories shown. The full geographic domain of the Arctic should be considered in the source region categorization. This is important since Lines 297-298 state that "The cumulative contribution of all regions outside the Arctic dominated air mass

[Figure]

history within the lower troposphere, making up to 97% of the PES", but this is likely partially due to the definition of "Arctic" in this work. For example, if the Beaufort and Chukchi Seas were included in the "Arctic open water" in Figure 8, would the trend be even stronger?

Figure 4b: What do the areas "not categorized in the selected geographic regions" correspond to? At >3 km, this corresponds to >20% of the data, and yet Figure 2 shows that all of the Northern Hemisphere north of 25 deg, except northern Africa, should be included in the source sector categorization. I'm also concerned about the use of 15 days of backward trajectories in terms of the associated spatial uncertainties in these trajectories, which I couldn't find to be addressed. Have the authors done analysis to validate or assess the uncertainty in the trajectories?

Additional comments: Intro, Lines 58-61: This sentence states that the references provided are "recent studies demonstrat[ing] gas flaring and shipping", but several of these studies examined oil and gas extraction activities and not specifically flaring. Further, the cited Winiger et al 2017 (PNAS) study shows flaring to be a minor BC source, and the more recent Winiger et al 2019 (Sci Adv, not currently cited in this work) also showed that the BC isotopes were not consistent with flaring. Therefore, this sentence needs to be revised to more accurately reflect the results in the manuscripts cited.

Intro, Lines 64-66: A missing and important summertime Arctic airborne study is ARCTAS-B campaign, which included flights over the Canadian Arctic during July 2008. See Matsui et al (2011a&b, JGR), which also included SP2 BC data and air mass source analysis, making it well-suited to also be considered by the authors for comparisons to the results herein.

Intro, Lines 66-68: Please consider revising this sentence. The ground-based studies can report the influence of long-range transport on the BL composition, but the strength of this work is the examination of the varying influence across the vertical column.

Methods, Lines 156-157: This needs to be rephrased because scaling to number concentrations allows quantitation but it does not "allow for assigning particle types to different sources" as stated here, as chemical composition and air mass trajectory analysis, not the concentrations themselves, are used to assign sources.

Figure 4 caption: Please add the dates of the Arctic and southern air mass periods in the caption for clarify and also refer the reader in the caption to Figure S1 for flight maps for context.

Figure 5: Please clarify whether an air mass can correspond to more than one source sector, as this seems quite possible. Please also clarify what is meant by: "The PES above vegetation fires refers to the 1-5 km vertical range." Are air masses than pass within the BL above a fire included? This is confusing as worded.

Lines 278-279: Based on Figure 4a, it appears that Siberia should be the largest contributor at >3 km, but that isn't mentioned here.

Lines 297-298: Figure 4b shows 30% Arctic in the BL, which seems to contradict the 97% outside of the Arctic quoted in this sentence.

Lines 303-304: It is stated that "the contribution of air masses that had resided above Arctic open water was significantly lower during the southern air mass period (Fig. 5b), compared to the Arctic air mass period." However, it looks pretty similar at <200 m in Fig 5b.

Line 324: Note that the mass spectra show oxalate, malonate, and succinate, rather than oxalic acid, malonic acid, and succinic acid (i.e. paired cation is not definitively known through the mass spectra – or at least it isn't discussed). Please use the appropriate wording throughout.

Line 325: Clarify here and elsewhere that you are referring to number percentages, since mass fractions are more commonly reported in the literature. Also, clarify whether 86% here refers to oxalate being present in 86% of the DCA-containing particles, as

this is not clear as stated here, and please clarify similar wording throughout.

Line 328 & Fig S8: To aid in the interpretation of this sentence, please label chloride and nitrate in this figure.

Figure 8: This figure and other similar figures show really nice analysis!

Lines 350-351: This sentence refers to air residence time but I don't see time in Figure 8, so I can't see how to evaluate this statement.

Lines 377: This statement is strange because the previous ground-based studies weren't sampling the FT, so this seems to be an unnecessary statement. It can either be removed, or the authors can simply refer to the ground-based data as being less influenced by long-range transport, whereas this work sampled both the BL and FT, which is a strength.

Line 427: This reference seems to be missing from the reference list.

SI Section 1. Definite hit rate for the non-single-particle mass spec reader.

Fig S8: Please clarify whether this statement means that 60%, by number of the sea spray aerosol particles containing at least one DCA, as the statement is confusing as written.

Figures S11 and S14: Add missing a and b labels to figure to agree with caption.

Figure S18: This seems to show that DCA-containing particles were not observed for the southern air masses. Is that correct?

SI Equation 5 (page 24): I believe there is a typo here, as the definition of uncertainty for binomial statistics should be: sigma = sqrt((PF(1-PF))/N). If this typo impacts the data uncertainties shown, then please fix throughout.

---

## Author Comment (AC1) · 15 Jan 2021

We thank Referee #1 for her/his comments and suggestions, which helped to improve the manuscript.

Our response is formatted as follows:

**Reviewer's comments**
Author's reply
Changes to the manuscript.

All page, line, section and figure numbers in bold refer to the original manuscript, all others to the revised version.

**Köllner et al. provide a detailed summary on Arctic boundary layer and lower free tropospheric aerosol composition during the summer using multiple analytical methods, satellite data, and air mass trajectory analyses. Two main air sectors were defined during their observations including Arctic air from the north and southerly transport from lower latitudes. Sources reported include wildfire, industrial, and other combustion sources in addition to marine sources such as sea spray. While this detailed information fills a key gap in aerosol observations during the Arctic summertime, there are a few issues that should be addressed prior to publication as delineated below.**

**The abstract is essentially a list of findings and contains no information on how the current work ties into broader implications for Arctic climate or why these results are novel. The authors should consider adding a couple sentences to demonstrate the importance of their work.**

We added the following sentences in the abstract:

"Aerosol particles impact the Arctic climate system, both directly and indirectly by modifying cloud properties, yet our understanding of their vertical distribution, chemical composition, mixing state, and sources in the summertime Arctic is incomplete. In-situ vertical observations of particle properties in the high Arctic, combined with modeling analysis on source attribution are in short supply, particularly during summer."
[…]
"Our findings improve our knowledge of mid-latitude and Arctic regional sources that influence the vertical distribution of particle chemical composition and mixing state in the Arctic summer."

**The authors provide a very comprehensive, detailed account of their results, but how their findings fit into the bigger picture is not apparent. A synopsis of why assessing the aerosol composition to this level of detail is needed to provide a clear picture on the importance of such measurements. For example, why do we care about knowing the abundance of DCA, sulfate, BC, and organic aerosols, specifically? What has previous work elucidated in terms of these specific aerosols and their impacts on radiation and cloud formation? What observations exist and why are the limited? What are models lacking that require data such as those from the current work? These are key questions for Arctic aerosol-cloud-radiation**

**studies that should be elaborated upon to motivate the purpose of this work. Surely, any Arctic aerosol scientist would know why this work is important; however, to emphasize the importance to a larger crowd (and, say, a cloud physicist looking for information on why aerosols are important for clouds), a broader context and implications should be included in the introduction and revisited in the conclusions.**

We re-formulated and re-structured the introduction, such that the impact of particle composition and mixing state on aerosol-radiation (ARI) and aerosol-cloud interactions (ACI) is discussed:

[revised manuscript text omitted]

**While the authors do compare their findings a couple of times to Schmale et al. (2011) and reference others like Shaw et al. (2010), Kawamura et al. (2012), and Leaitch et al. (2018), the comparisons are quite limited, even to the extent where they state "This finding is consistent with previous ground-based and shipborne studies…" and only provide references but not what the findings from those studies. In order to put their results into the context of others, and demonstrate why their results fill key observational gaps, a mentioning of how their results compare to other analogous studies by briefly describing what those studies concluded is needed. Also, other key studies such as those by Quinn et al. (2002, 2009) and Winiger et al. (2019), to name a few, that describe aerosol composition at various sites throughout the Arctic should be included in the discussion and comparison of results.**

Studies by Quinn et al. (2002) and Winiger et al. (2019) were added in the Introduction and Discussion as follows:

Introduction:

[revised manuscript text omitted]

Discussion Sect. 3.3.2:

"Earlier measurements show that background concentrations of ammonium and nitrate are generally low in Arctic summer (Quinn et al., 2002; Kuhn et al., 2010; Chang et al., 2011; Schmale et al., 2011; Quennehen et al., 2011; Hamacher-Barth et al., 2016; Lange et al., 2018), which we can confirm under the pristine conditions during the Arctic air mass period (Fig. 10). However, our vertical profile measurements during the southern air mass period (Fig. 15) show that episodic and localized transport of vegetation fire emissions can perturb background concentrations of nitrate, ammonium, and organic matter. This finding is in line with modeling and observational studies (Hecobian et al., 2011; Brock et al., 2011; Lathem et al., 2013; Kondo et al., 2011; Matsui et al., 2011a; Breider et al., 2014), reporting on the response of nitrate, ammonium, and organic concentrations in the Arctic on the transport of large fire emissions from sub-Arctic regions. Also gas-phase measurements demonstrate the large input of ammonia (precursor gas of particulate ammonium) by vegetation fire emissions to the summertime Arctic lower troposphere (Lutsch et al., 2016)."

"Consistently, it has previously been shown that the sulfate burden in the summertime Arctic FT is dominated by transport of a wide variety of anthropogenic emissions in northern Eurasia, North America, and East Asia (e.g., Shindell et al., 2008; Hirdman et al., 2010; Kuhn et al., 2010; Bourgeois and Bey, 2011; Schmale et al., 2011; Matsui et al., 2011a; Breider et al., 2014; Yang et al., 2018; Sobhani et al., 2018)."

**Specifically, for the air mass source assessment, a more direct link to Stohl (2006) is warranted, given that study evaluated air mass sources of the Arctic using FLEXPART in detail. Are the current results consistent or contradictory to previous work?**

We added more detailed comparisons to Stohl (2006) and Klonecki (2003) in Sections 3.2.1 and 3.3.1:

[…]
"Air mass history during the first period reflects the concept of isentropic transport to Arctic regions during summer. **Klonecki et al. (2003) and Stohl (2006)** provide comprehensive analyses on the role of transport pathways into the Arctic troposphere across seasons. It was found that the Arctic summertime lower troposphere is quite isolated from southern latitude sources as diabatic low-level transport into the polar dome is largely absent. This is in line with our results. We found that the near-surface regions are largely isolated from the rest of the atmosphere, whereas regions aloft are episodically influenced by air originating from southern latitudes (Fig. 7a)."
[…]
"However, contributions from southern latitude regions, i.e. Europe, Siberia, northern Canada, Greenland, and the Atlantic Ocean, increased with altitude due to enhanced quasi-isentropic transport from these regions into the high Arctic (Fig. 7a). The abovementioned studies by **Stohl (2006) and Klonecki et al. (2003)** demonstrate that emissions in Siberia, Europa, and Asia can influence Arctic summertime composition in altitudes above the BL, which is consistent with our results."
[…]
"This period is referred to as southern air mass period, since air mass history shows the prevalence of air masses originating from southern latitudes. Lagrangian air mass history analysis suggests a pronounced impact of southern latitude sources on Arctic composition (Fig. 14). The cumulative contribution of all regions outside the Arctic dominated air mass history within the lower troposphere (Fig. 14a). By comparing the synoptic situation during the southern air mass period with climatological mean (see Supplement Sect. 3), we found that the presence of the low-pressure system led to a significantly anomalous synoptic situation. This finding explains the discrepancy between our results on air mass history during the second period and results on Arctic transport climatology by **Stohl (2006)**, revealing the largely unperturbed Arctic lower troposphere during summer."

**The results and discussion section seems a bit fragmented as the discussion goes back and forth between the north and south influences, and describing single particle composition for both is combined. Perhaps the entire section would be easier to follow if all data (composition, FLEXPART, and satellite data) were discussed in tandem but organized by source region (north versus south).**

Thanks for the helpful suggestion. We re-structured the "Results and discussions" section as follows:

3.1 Single particle chemical composition
3.2 Arctic air mass period
    3.2.1 Meteorological overview and air mass history
    3.2.2 Arctic marine influences on particle composition
    3.2.3 Long range transport influences on particle composition
3.3 Southern air mass period
    3.3.1 Meteorological overview and air mass history
    3.3.2 Vegetation fire and anthropogenic influences on particle composition

**Also, why are vegetation fires and anthropogenic sources a separate section? Aren't they technically a south air mass influence sources and aren't they technically long-range transported? Not clear why these are segregated from the section on the southerly sources.**

The section 3.3.2 now includes both the discussion on source regions and sectors during the southern air mass period.

**Seems like Figure S1 should be in the paper. There is a description of the site locations in text, but it is not clear to those unfamiliar with the area. I would think this is important to show for a clear link between sources and transport pathways, and the type of land they cover.**

The figure has been added to the main manuscript.

**It was not clear to me until later in the methods why the UHSAS size range was restricted. Please provide the size range of the ALABAMA in section 2.2.1. Also, what is the size range of the SP2 data used?**

The ALABAMA size range is given in Sect. 2.2.1. To make it more clear, we added the following sentence in Sect. 2.3:

"For the conversion of the ALABAMA fraction into number concentration (see Sect. 2.4), UHSAS number concentrations in a size range between 320 nm and 870 nm ($N_{>320}$) are used (see Sect. 2.2.1). "

The size range of the SP2 had been added in Sect. 2.2.3.

**The statement on the top of page 16 indicates the aerosol data presented are from "sampling outside of clouds". But does that mean the measurements were conducted during cloud free periods or were the aerosols sampled below/above clouds during cloudy periods, or some combination of each? I assume based on the generalized north and south air mass descriptions that clouds were mostly present during the latter half of the study, but it is not clear when exactly the aerosols were sampled during cloudy versus clear conditions. This would be important to discuss, at least briefly, because aerosol composition can be quite different in the Arctic boundary layer during clear versus cloudy conditions.**

We analyzed measurements that were conducted outside clouds, irrespective if clouds were present or absent. The selection "sampled outside clouds" was applied because the aerosol inlet was not suitable for in-cloud sampling. Thus, measurements inside clouds were discarded by using data from an under-wing FSSP (Forward Scattering Spectrometer) probe (Leaitch et al. (2016)).

It is correct. The Arctic air mass period was characterized by generally clear skies with occasionally broken stratocumulus clouds, whereas the southern air mass period was characterized by overcast sky (see Sections 3.2.1 and 3.3.1). Further details on the presence and properties of clouds during the NETCARE 2014 airborne study can be found in Leaitch et al. (2016).

However, given that air mass history changed during the course of the study and at the same time we observed different cloud characteristics, it is difficult to study solely the interaction between aerosol composition and clouds. Further, the ALABMA particle counting statistics during the NETCARE 2014 study is not sufficient to conduct a cloud-by-cloud analysis.

During the Arctic airborne campaign ACLOUD in summer 2017, the ALABAMA was operated with an improved setup providing a better counting statistic (Clemen et al., 2020, AMT). In addition, a counterflow virtual impactor was used to sample and analyze cloud residuals. We will thus announce an upcoming publication (Eppers et al, in prep.) that will focus on the coupling between aerosol and clouds during this campaign.

**What is the relative contribution of the Arctic versus southern air mass periods? It is obvious for the beginning and end, but what about the 5-day transitional period? The authors could provide the % of each air mass contribution for the entire study at the bottom of page 11.**

We added the following information in Sect. 3.2:

"The synoptic conditions changed over the course of NETCARE 2014 from an initial Arctic air mass period (July 4-12, **~26 measurement hours**) to a southern air mass period (July 17-21, **~19 measurement hours**) with a transition in between (Burkart et al., 2017; Bozem et al., 2019)."

We did not consider to study the air mass history of the transitional period, because flying was impeded during the transitional period. Thus, the focus of this paper is on the Arctic and southern air mass periods, which provide airborne measurements on particle chemical composition.

**Why is organic:sulfate only shown in select panels/figures (i.e. Figures 10, 11, 14, and 15)? The ratio is also shown as a different color in Figure 11a but should be consistent with the rest of the figures.**

We had chosen different colors for different parameters. Dark green refers to the organic-to-sulfate ratio. Pink refers to the sulfate-to-organic ratio.

The organic-to-sulfate ratio as a function of altitude and as a function of residence time over Arctic open water was presented in Willis et al. (2017). To differentiate from our previous work, we will not show these results again.

For consistency, we added the sulfate-to-organic ratio in Fig. 10, Fig. 13b, and Figs. 17b and c to analyze the anthropogenic influence on particle composition, according to Schmale et al. (2011). In Fig. 18b, we added the organic-to-sulfate ratio to analyze the influence of biomass burning on particle composition. The new results and trends are discussed in the respective sections.

**For Figure 15, there does not seem to be major differences between the PES sources for fires, population, and industry. The authors describe each separately, but some discussion on why they look so similar is needed.**

We re-scaled the y-axes to better view the differences. However, we agree, trends look similar, but differentiation in the absolute values can be seen. For example, highest sulfate concentrations were observed in the Arctic at maximum PES fraction over anthropogenic sources. Further, the organic-to-sulfate ratio was increasing with vegetation fires PES fraction, which is not the case for anthropogenic sources (compare with sulfate-to-organic ratio). We added these points in Sect. 3.3.2.

Similar trends for nitrate, sulfate, organics, and ammonium might be caused by different reasons. First, precursor gases (such as $NO_x$, $SO_4$, $NH_3$, VOCs) are emitted by anthropogenic sources and vegetation fires on either side. Second, during long-rang transport (irrespective of the source region/sector), we would expect that particles are exposed to intensive atmospheric aging processes, which predominantly affect particle composition. Third, different source sectors can contribute to air mass history 15 days prior to sampling (see comment and reply Reviewer#2 and Supplement Sect. 9). As a result, industrial sources cannot be differentiated from populated areas, partly due to their close proximity.

[revised manuscript text omitted]

---

## Author Comment (AC2) · 15 Jan 2021

We thank Referee #2 for her/his comments and suggestions, which helped to improve the manuscript.

Our response is formatted as follows:

**Reviewer's comments**
Author's reply
Changes to the manuscript.

All page, line, section and figure numbers in bold refer to the original manuscript, all others to the revised version.

**Kollner et al present summertime aircraft-based online aerosol chemical composition data from two mass spectrometers, providing both single-particle and bulk composition information, which is quite unique. These novel data were collected as part of the Canadian NETCARE campaign and significantly contribute to understanding of Arctic summertime aerosol composition, knowledge of which is observationally limited, especially when considering non-ground-based measurements. This is an important dataset to publish. Overall, the paper is well-done and contributes significantly to the field. The vertical profiles of aerosol composition and source region analyses using backward air mass trajectories are a good analysis approach to correlate composition with air mass source regions. It is excellent that the authors scaled the ALABAMA data to number concentrations and conducted uncertainty analysis. The supplementary material is comprehensive.**

**However, I have one major data analysis concern, as described below, about the significant fraction of unexamined single-particle data, the omission of which potentially impacts the presentation and interpretation of the results.**
**My major data analysis concern regards the approach in the classification of the ALABAMA single-particle mass spectra. As stated on lines 140-141, "the ion marker method classified particles based on the presence of pre-selected species that are of interest." This means that the authors have chosen what chemical species to focus on, which is not necessarily problematic; however, only 54% of the single-particle mass spectra were classified. This means, as shown in Figure 6 and stated in the Figure S9 caption and on line 320, that 46% of the single-particle mass spectra are not included in the discussed data/results. Being nearly half of the ALABAMA data and the fact that this paper is meant to be an overview/summary of the results during NETCARE, I am concerned that the authors may inadvertently be biasing their results by not classifying the remaining data, and I do not consider 46% to be an acceptable fraction of unclassified/ real spectra. Figure S9 shows that at least the majority of these mass spectra are real, and therefore, they need to be classified and discussed. Figure 6 shows that this requires further ion marker searches to classify the particles, and a good starting point for this is the examination of the mass spectra shown in Figure S9. In their previous paper (Kollner et al 2017, ACP), which focused on trimethylamine-containing particles, they included an additional particle type called "K-containing" (potassium-containing). Figure S9 shows the mean mass spectrum of the "Others" (non-classified) particles, and this shows a dominant K+ peak, carbonaceous ions, and sulfate. Figure S9 is also nearly identical to the mean K-containing mass spectra in Figure 4e of Kollner et al (2017, ACP). Since this particle type is consistent with biomass burning (the mass spectrum of which is discussed on lines 441-444), it seems even more important to include a K-**

containing, or similar, particle type to classify more of the remaining data. Given the similarity of Figure S9 to biomass burning, it is necessary for these data to be included in the evaluation of the vegetation fire influence portion of the manuscript in particular. How does the PF.N>320 vs vegetation fire PES fraction compare for when all K-containing particles are included, or how does it compare currently for this "Others" category as a starting point? Conducting a K+ search on the remaining mass spectra will also be informative to then examine the mean mass spectra of the still unclassified particles, which can be further classified to potentially unearth another particle type that may provide greater insights. It is possible that by including more of the available data that some of the presented trends may change, or new trends may emerge, and without including more of the available data, I am concerned about the results being biased by the chosen ion markers.

> Potassium was included in more than 60 % of all particles analyzed during the NETCARE 2014 measurements. Moreover, potassium is thus largely internally mixed with other substances, such as TMA, nitrate, and DCA (see modified Fig. 5), making it difficult to draw conclusions on the source regions/sectors of this substance. However, with 22 %, potassium-containing particles make up a large fraction from the group of "Others". We therefore decided to work with this K-containing sub-type (22 %). The analysis of this particle type is presented in Figs. 5, 6f, 10, 12, 13, S11, and S14. The new results are discussed in the respective sections.

> Further, we classified the rest (24% of all particles) based on marker species ammonium, methanesulfonic acid, and sulfate and show this classification in the Supplement Fig. S8. By this analysis, the reader can comprehend the composition of the remaining particles, even though these particle sub-groups are not further analyzed. Having done this, 16 % of all particles still remain unclassified. The mean spectrum of "Others" thus changed and is shown in the Supplement Fig. S9. We added a sentence in Sect. 3.1, Lines 315-317:

> "24% of all mass spectra are not considered for the further analysis (gray filling in Fig. 5), however, those remaining mass spectra are further sub-classified with marker species sulfate, ammonium, MSA etc. (see Figs. S8 and S9)."

**Section 2.5.2 & Figure 3: Why does the category of "Arctic open water" not include the full Arctic Ocean, including the Beaufort and Chukchi Seas? Figure S36 shows some surface influence from this region, so inclusion of only open water north of 73.5 deg means that additional Arctic open water is instead classified as "Alaska" and missed in the "Arctic open water" source sector analysis. The authors could sub-categorize the High Arctic, but the full Arctic Ocean should be considered as "Arctic" for consistency with its geographic definition and for comparison to other work.**

**In addition, as described below, the authors' definition of "Arctic" as north of 73.5 deg is inconsistent with its geographic definition and norms of other studies, and impacts the source region attribution and discussion. Otherwise, the majority of my other comments are minor.**
**This is also true with respect to the definition of 73.5 deg north as "Arctic", when the Arctic Circle is at 66 deg 34' N. Given little population north of the Arctic Circle, it would seem more appropriate to either sub-divide the Arctic into two categories, or include 66-73 N in the Arctic category. The categories of Alaska, northern Canada, Europe, Greenland & the**

**Atlantic, and Siberia extend much, much further north, and therefore, air mass influence still in the "Arctic" (i.e. >66 deg N) would be discussed as being from a further south influence with the categories shown. The full geographic domain of the Arctic should be considered in the source region categorization. This is important since Lines 297-298 state that "The cumulative contribution of all regions outside the Arctic dominated air mass history within the lower troposphere, making up to 97% of the PES", but this is likely partially due to the definition of "Arctic" in this work. For example, if the Beaufort and Chukchi Seas were included in the "Arctic open water" in Figure 8, would the trend be even stronger?**

We do not agree that the usage of the word *Arctic* when discussing transport related atmospheric and meteorological processes is being fixed or defined by the definition of the Arctic circle (> 66° N). Earlier studies demonstrated that the Arctic lower troposphere (LT) is characterized by upward-sloping isentropes that form a dome-like structure (Carlson, 1981; Iverson, 1984; Barrie, 1986). This structure later became known as the polar dome (Klonecki et al., 2003; Law and Stohl, 2007). The location of the polar dome boundary, and thereby the transport barrier that isolates the Arctic LT from lower latitudes, is often characterized by the location of the Arctic front (Klonecki et al., 2003; Law and Stohl, 2007).

In our recent study (Bozem et al., 2019, ACP), we applied measurements of trace gas gradients for the identification of the polar dome boundaries. As a result, the polar dome boundary during the NETCARE 2014 campaign was located at around 73.5° N, which is in agreement with a recent study by Crawford and Serreze (2015), demonstrating the more northern location of the Arctic front in summer. However, the location is variable in space and time and thus needs individual analysis for different measurement campaigns. We argue that the definition of the polar dome latitudinal boundary is more appropriate to use when studying atmospheric processes in high latitude regions. Along with this, there is no consensus in literature about the definition of Arctic regions or a convention to define Arctic regions in atmospheric studies. As examples: Liu et al. (2015) and Stohl (2006) defined the Arctic as being north of 70° N; Shindell et al. (2008) used a definition of 68° N; Yang et al. (2018) and Xu et al. (2017) described the Arctic as being north of 66.5° N, whereas Zhu et al. (2020) and Stohl et al. (2013) used > 66° N.

We added Table 2, specifying the latitudinal and longitudinal boundary of the source regions. This table should provide a better overview for the reader and show clearly show that the method used here to define Arctic region is specifically applied for the July 2014 measurements.

**Figure 4b: What do the areas "not categorized in the selected geographic regions" correspond to? At >3 km, this corresponds to >20% of the data, and yet Figure 2 shows that all of the Northern Hemisphere north of 25 deg, except northern Africa, should be included in the source sector categorization.**

"Not categorized" corresponds to regions in Mexico and northern Africa, as given in Fig. 3. For the second period, data on aerosol composition above 3 km are sparse. Thus, the further data analysis will not consider altitudes above 3 km.

We added the following sentences in Sect. 3.3.1:

"Regions that were not categorized (Mexico and northern Africa both north of 25° N, see Fig. 3) contribute significantly to air mass history at altitudes above 3 km (Fig. 14). However, our data on aerosol composition at altitudes above 3 km are sparse, with the result that our further data analysis is limited to altitudes below 3 km."

**I'm also concerned about the use of 15 days of backward trajectories in terms of the associated spatial uncertainties in these trajectories, which I couldn't find to be addressed. Have the authors done analysis to validate or assess the uncertainty in the trajectories?**

We did not perform a dedicated uncertainty analysis, in terms of analysing the variability given by the meteorology, e.g., through using meteorological ensemble data. However, our approach provides a certain uncertainty analysis with respect to the release of the individual tracer particles. In total, we released an ensemble of 20,000 individual air particles (i.e., trajectories) in each 10-min flight interval within a 3-dimensional air volume with maximum and minimum boundaries in space given by the flight track. Thus, we do not analyse individual trajectories, but the averaged (correct: statistically anticipated) value of 20,000 individual calculations, which together provide a certain probability distribution.

Of course, 15 days is a long time period where many bifurcation points can alter the path of the trajectories. However, this is partially considered in the analysis by the vast ensemble of trajectories. More so, prior Arctic studies give us confidence that our analysis, using 15-days backward trajectories, is well suited to study the Arctic summertime composition with respect to transport processes (e.g., Hirdmann et al., 2010; Stohl, 2006; Stohl et al., 2007)

**Additional comments: Intro, Lines 58-61: This sentence states that the references provided are "recent studies demonstrat[ing] gas flaring and shipping", but several of these studies examined oil and gas extraction activities and not specifically flaring.**

We updated the list of references and separated references including results on oil/gas extraction and flaring:

"Regarding high-latitude anthropogenic sources, recent studies demonstrated oil/gas extraction activity and shipping to significantly impact the lower tropospheric BC, organic, and sulfate aerosol burdens (e.g., AMAP, 2010; Eckhardt et al., 2013; Breider et al., 2014; Ferrero et al., 2016; Gunsch et al., 2017; Creamean et al., 2018; Kirpes et al., 2018). The contribution of high-latitude flaring emissions to Arctic BC concentration is controversially discussed. While some studies suggest gas flaring to be an important source of BC, particularly in winter and spring (Stohl et al., 2013; Xu et al., 2017; Leaitch et al., 2018; Zhu et al., 2020), others provide evidence that flaring plays a minor role (Winiger et al., 2017, 2019)."

**Further, the cited Winiger et al 2017 (PNAS) study shows flaring to be a minor BC source, and the more recent Winiger et al 2019 (Sci Adv, not currently cited in this work) also showed that the BC isotopes were not consistent with flaring. Therefore, this sentence needs to be revised to more accurately reflect the results in the manuscripts cited.**

As mentioned above, we modified the sentences according to your suggestion as follows:

"The contribution of high-latitude flaring emissions to Arctic BC concentration is controversially discussed. While some studies suggest gas flaring to be an important source of BC, particularly in winter and spring (Stohl et al., 2013; Xu et al., 2017; Leaitch et al., 2018, Zhu et al., 2020), others provide evidence that flaring plays a minor role (**Winiger et al., 2017, 2019**)."

**Intro, Lines 64-66: A missing and important summertime Arctic airborne study is ARCTAS-B campaign, which included flights over the Canadian Arctic during July 2008. See Matsui et al (2011a&b, JGR), which also included SP2 BC data and air mass source analysis, making it well-suited to also be considered by the authors for comparisons to the results herein.**

We added the references Matsui et al. (2011a,b) in the Introduction and the Discussion as follows:

Introduction:

"Boreal fires and to a lesser extent anthropogenic activities in North America and northern Eurasia can strongly influence the organic aerosol burden in the summer Arctic free troposphere (FT) (Hirdman et al., 2010; Schmale et al., 2011; **Matsui et al., 2011a;** Lathem et al., 2013; Breider et al., 2014)."

"Sulfate concentrations in the summertime Arctic FT are largely influenced by anthropogenic sources in northern Eurasia, North America, and East Asia (Shindell et al., 2008; Hirdman et al., 2010; Kuhn et al., 2010; Bourgeois and Bey, 2011; Schmale et al., 2011; **Matsui et al., 2011a;** Breider et al., 2014; Yang et al., 2018; Sobhani et al., 2018)."

"In particular, airborne studies that attribute aerosol physical and chemical properties to sources are sparse, especially in summer (Radke and Hobbs, 1989; Brock et al., 1989; Paris et al., 2009; Schmale et al., 2011; Quennehen et al., 2011; **Matsui et al., 2011a, b;** Kupiszewski et al., 2013; Creamean et al., 2018)."

Discussion Sect. 3.2.3:

"This is in agreement with previous Arctic measurements of BC (or EC) at different seasons (e.g**., Matsui et al., 2011b**; Winiger et al., 2019), demonstrating particularly low concentrations during summer compared to the rest of the year. **Matsui et al. (2011b)** could show the influence of Asian anthropogenic emissions on the BC concentration in Arctic summer, in line with our results during the Arctic air mass period."

"It has previously been demonstrated that anthropogenic emissions in northern Eurasia and East Asia contribute to enhanced sulfate concentrations in the summertime Arctic FT (e.g., **Matsui et al., 2011a;** Breider et al., 2014; Yang et al., 2018). "

Discussion Sect. 3.3.2:

"It is thus conceivable that this transport pathway was linked to intensive particle wash-out events (Garrett et al., 2010, 2011; **Matsui et al., 2011a**; Browse et al., 2012; Sato et al., 2016)."

"This finding is in line with modeling and observational studies (Hecobian et al., 2011; Brock et al., 2011; Lathem et al., 2013; Kondo et al., 2011; **Matsui et al., 2011a**; Breider et al., 2014), reporting on the response of nitrate, ammonium, and organic concentrations in the Arctic on the transport of large fire emissions from sub-Arctic regions."

"Consistently, it has previously been shown that the sulfate burden in the summertime Arctic FT is dominated by transport of a wide variety of anthropogenic emissions in northern Eurasia, North America, and East Asia (e.g., Shindell et al., 2008; Hirdman et al., 2010; Kuhn et al., 2010; Bourgeois and Bey, 2011; Schmale et al., 2011; **Matsui et al., 2011a**; Breider et al., 2014; Yang et al., 2018; Sobhani et al., 2018)."

**Intro, Lines 66-68: Please consider revising this sentence. The ground-based studies can report the influence of long-range transport on the BL composition, but the strength of this work is the examination of the varying influence across the vertical column.**

This sentence had been removed.

**Methods, Lines 156-157: This needs to be rephrased because scaling to number concentrations allows quantitation but it does not "allow for assigning particle types to different sources" as stated here, as chemical composition and air mass trajectory analysis, not the concentrations themselves, are used to assign sources.**

We modified this sentence as follows:

"In the following, we present the conversion of unscaled ALABAMA measurements into quantitative particle number concentrations."

**Figure 4 caption: Please add the dates of the Arctic and southern air mass periods in the caption for clarify and also refer the reader in the caption to Figure S1 for flight maps for context.**

Done.

**Figure 5: Please clarify whether an air mass can correspond to more than one source sector, as this seems quite possible.**

Yes, air mass history is typically influenced by different sources; in particular, if source sectors are in close proximity. We added Section 9 in the Supplement to show comparisons of PES fractions between different source sectors for the first period (Figs. S37a-c) and for the second period (Figs. S37d-f). It is obvious that air masses with high Arctic open water PES fraction are largely isolated from other sources (Fig. S37a); whereas anthropogenic sources (industrial and populated areas) often contribute on either side to air mass history (Figs. S37 c and f). However, a pre-processing of the PES fractions was applied if possible, in order to differentiate different source sector contributions from each other.

We added the following sentences in Sect. 2.5.2:

"Different source sectors can contribute to air mass history within the 15 days prior to sampling. The Supplement Sect. 9 shows comparisons of PES fractions between different source sectors. It was found that air masses with high Arctic open water PES fraction are largely isolated from other sources; whereas anthropogenic sources (industrial and populated areas) can contribute on either side to air mass history by their close proximity. However, a pre-processing of the PES fractions was applied, if possible, in order to differentiate different source sector contributions from each other. For example, we could separate contributions of vegetation fires to air mass history from anthropogenic sources. Further details can be found in the Supplement Sect. 9."

**Please also clarify what is meant by: "The PES above vegetation fires refers to the 1-5 km vertical range." Are air masses than pass within the BL above a fire included? This is confusing as worded.**

Air masses that were injected between 1 and 5 km (so-called footprint layer) above vegetation fires are included in the analysis. Injection heights of vegetation fires typically vary with fuel type, temperature etc. Several studies show injection heights for boreal fires typically between 1 and 5 km. We discussed this topic briefly in Sect. 2.5.1 and in more detail in the supplementary material (Sect. 2.1).

The caption of Figs. 7 and 14 had been changed as follows:

"The FLEXPART derived contribution is expressed as a fraction of the potential emission sensitivity (PES) in the model domain lowest vertical level (0 – 400 m, except for vegetation fires) over a 15-days backward simulation. A vertical injection layer between 1 and 5 km is applied for vegetation fires (see details in Sect. 2.5.1)."

**Lines 278-279: Based on Figure 4a, it appears that Siberia should be the largest contributor at >3 km, but that isn't mentioned here.**

We added the following sentence:

"The contribution of Siberian regions to air mass history is highest in altitudes above 3 km."

**Lines 297-298: Figure 4b shows 30% Arctic in the BL, which seems to contradict the 97% outside of the Arctic quoted in this sentence.**

We agree. This is confusing as worded. We removed the part "making up to 97% of the PES".

**Lines 303-304: It is stated that "the contribution of air masses that had resided above Arctic open water was significantly lower during the southern air mass period (Fig. 5b), compared to the Arctic air mass period." However, it looks pretty similar at <200 m in Fig 5b.**

We changed the sentence as follows:

"Also, the contribution of air masses in the lower FT that had resided above Arctic open water was significantly lower during the southern air mass period above (compare Figs. 7b and 14b), compared to the Arctic air mass period."

**Line 324: Note that the mass spectra show oxalate, malonate, and succinate, rather than oxalic acid, malonic acid, and succinic acid (i.e. paired cation is not definitively known through the mass spectra – or at least it isn't discussed). Please use the appropriate wording throughout.**

The ALABAMA mass spectra typically show ion fragments of the analyzed particles, owing to the high energy laser-ablation process. We thus use laboratory measurements to analyze the fragmentation pattern of atmospheric compounds. The laboratory study by Silva and Prather (2000) demonstrated that mass spectra of carboxylic acids show deprotonated negative parent ions, such as oxalate for measured oxalic acid. We added the following sentence for clarification:

"It should be noted that the ALABAMA detects oxalate-, malonate-, and succinate ions that most likely originate from oxalic, malonic, and succinic acid, respectively (Silva and Prather, 2000)."

**Line 325: Clarify here and elsewhere that you are referring to number percentages, since mass fractions are more commonly reported in the literature.**

We added the following two sentences:

Lines 206-207: "To note, the following use of the word *fraction* always refers to the number fraction measured by the ALABAMA."

Lines 314-315: "To note, percentages given in this study always refer to number percentages measured by the ALABAMA."

**Also, clarify whether 86% here refers to oxalate being present in 86% of the DCA-containing particles, as this is not clear as stated here, and please clarify similar wording throughout.**

This was changed as follows:

"The term particulate DCA implies the presence of oxalic, malonic, and/or succinic acid (see Table 1) with oxalic acid as most abundant (in 86 % of DCA particles), followed by succinic acid with 41 % of DCA particles and malonic acid with 38 % of all DCA particles (not shown)."

**Line 328 & Fig S8: To aid in the interpretation of this sentence, please label chloride and nitrate in this figure.**

Done.

**Figure 8: This figure and other similar figures show really nice analysis!**

Thanks.

**Lines 350-351: This sentence refers to air residence time but I don't see time in Figure 8, so I can't see how to evaluate this statement.**

Thanks for making us aware of this mistake. All respective figures show the PES fraction on the x-axis. We modified this sentence as follows:

"Particulate TMA was predominantly abundant when the air resided for more than 50% of the 15 days (PES fraction) prior to sampling over Arctic open water areas (Fig. 8)."

All respective figure captions in the main document and supplementary part had been changed accordingly.

**Lines 377: This statement is strange because the previous ground-based studies weren't sampling the FT, so this seems to be an unnecessary statement. It can either be removed, or the authors can simply refer to the ground-based data as being less influenced by long-range transport, whereas this work sampled both the BL and FT, which is a strength.**

The sentence had been deleted.

**Line 427: This reference seems to be missing from the reference list.**

The reference is listed in line 1164.

**SI Section 1. Definite hit rate for the non-single-particle mass spec reader.**

We added the following sentences in the SI:

"The instrument hit rate is defined as the number of particles that are successfully ionized by the ablation laser and that create a mass spectrum relative to the number of laser shots (Clemen et al., 2020). The triggered shot requires that the particle velocity was prior successfully determined and that the laser was ready to shoot (Brands et al., 2011). The maximum shot repetition rate of the Nd:YAG laser was set to 12 Hz."

**Fig S8: Please clarify whether this statement means that 60%, by number of the sea spray aerosol particles containing at least one DCA, as the statement is confusing as written.**

The caption had been modified as suggested:

"Expanded mean anion spectrum of 60 % (by number) of the sea spray particles containing at least one DCA."

**Figures S11 and S14: Add missing a and b labels to figure to agree with caption.**

Done.

**Figure S18: This seems to show that DCA-containing particles were not observed for the southern air masses. Is that correct?**

We see a clear vertical trend for DCA fraction during the first period. However, this is not the case for the second period. A reason might be that meteorological conditions during the second period led to more mixing between the BL and FT. We thus decided to discuss the DCA abundance in relation with the first period, where the separation between BL and FT air is clearer.

**SI Equation 5 (page 24): I believe there is a typo here, as the definition of uncertainty for binomial statistics should be: sigma = sqrt((PF(1-PF))/N). If this typo impacts the data uncertainties shown, then please fix throughout.**

The equation you noted above is the same as Eq. 5 in the SI, because $1/sqrt(N)=$ sqrt(N)/N.

[revised manuscript text omitted]

---

## Author Response (AR2)

Dear Editor Sally Ng,

we want to thank Referee #2 for her/his follow-up review, comments and suggestions, which helped to improve the manuscript.

Our response is formatted as follows:

**Reviewer's comments**
Author's reply
Changes to the manuscript.

All page, line, section and figure numbers in bold refer to the original manuscript, all others to the revised version.

**Kollner et al have made useful revisions to their manuscript overview of Arctic single-particle composition during NETCARE flights. In particular, the addition of the sub-potassium particle type (22% of the total) and application of their data analysis approach to these particles was a critical addition. Also, it is important that the composition of another 24% of the particles (ammonium, MSA, and sulfate) is acknowledged, leaving only 16% unclassified. This is an acceptable fraction of unclassified particles. Overall, the many revisions made to the manuscript have significantly improved the work, and these revisions have sufficiently addressed my previous comments and concerns. I only have a few remaining comments, mostly with respect to added text.**

Thank you.

**Lines 41-42: This new sentence is lacking a reference and is confusing as written, as it states that "the presence of aerosol in Arctic tropospheric layers leading to cooling of the surface beneath by absorbing incoming solar radiation and by reflecting radiation back to space." Certainly reflection would lead to cooling, but absorption should lead to warming. Please clarify sentence and add an appropriate reference.**

We added references and re-wrote the sentences as follows:

Lines 41-44: "Second, the presence of absorbing aerosol in Arctic tropospheric layers can lead to warming in the lower troposphere, but to cooling of the surface beneath by absorbing incoming solar radiation in this layer (e.g., Treffeisen et al., 2005; Engvall et al., 2009; Flanner, 2013). The result is an increase in tropospheric stability (e.g., Flanner, 2013)."

Lines 47-49: "In contrast to light-absorbing aerosol, scattering aerosol species, such as sulfate, of both anthropogenic and biogenic origin, exert a net negative shortwave radiative forcing on the Arctic surface by reflecting radiation back to space (Quinn et al., 2008; Yang et al., 2018)."

**Lines 53-55: Please add references for these new sentences.**

We added the following references:

Lines 53-59: "Aerosol particles, serving as nuclei for water condensation or nucleation of the ice phase, are fundamental to cloud formation (e.g., Lohmann and Feichter, 2005). The effects

of these particles on clouds are important drivers of the Arctic surface energy budget (e.g., Lubin and Vogelmann, 2006; Zamora et al., 2016). It is known that the net radiative effect of Arctic low-level clouds varies significantly with season (Intrieri et al., 2002; Shupe and Intrieri, 2004). Arctic low-level clouds warm the Arctic surface through most of the year. However, for a short period in summer when the incoming solar radiation is maximum over regions with a low albedo, clouds exert a negative radiative forcing on the Arctic surface (Intrieri et al., 2002; Shupe and Intrieri, 2004)."

**Lines 504-506: This new sentence states here that "the enhanced abundance of the sub-K particle type…with anthropogenic influence shows that potassium does need not necessarily be linked to a biomass burning source". This is in section 3.3.2 "Vegetation fire and anthropogenic influences on particle composition". While a subset of the sub-K particles were associated with anthropogenic influence rather than wildfires, this does not rule out biomass combustion as the source, rather it only rules out the satellite-detected wildfires as the source. Note that biomass burning particles are the most dominant particle type in the FT outside of the Arctic (see Schill et al. 2020, Nature Geosc.; Pratt and Prather 2010, JGR; Hudson et al. 2004, JGR). I suggest revising the current statement and linking to these previous papers.**

We have added the reference Schill et al. (2020) and re-formulated the sentences as follows:

Lines 505-508: "Given that the majority of nss-nitrate-containing particles were internally mixed with potassium (see Sect. 3.1), we have additional indications for their biomass burning origin (Silva et al., 1999; Hudson et al., 2004; Pratt and Prather, 2009; Pratt et al., 2011; Quinn et al., 2002; Schill et al., 2020). The inclusion of nitrate and sulfate with potassium is indicative for atmospheric processing of biomass burning particles while transported into the Arctic."

**If possible, I encourage moving the highly useful Fig S11 to the main text. In my opinion, Figures 2 and 4 could be moved to the SI.**

We moved Fig. S11 to the main manuscript (new Figs. 12 and 19). Figure 2 was moved into the Supplement (new Fig. S1). We accordingly changed the text in the main document and in the Supplement. We decided not to move Fig. 4 in the Supplement, since this is an essential graph to show the northern hemispheric distribution of the different source locations.

We further added a new reference (Schmale et al. (2021)) to the Introduction (line 116).

We found a few typos in Table 2 that were corrected and marked in the revised version.

Best regards
Franziska Köllner